# An Adversarial Training Approach to Robustify Stable Diffusion Systems Against Prompting Attacks

## Abstract

Text-to-Image (T2I) systems are generative models designed to generate images based on textual descriptions. Despite their remarkable performance, it has been shown that they are susceptible to misuse. One form of misuse involves manipulating the input prompt, leading to images that do not match with the given description. To address this, we introduce an adversarial training (AT) procedure for Stable Diffusion. Our aim is to train the model across various concepts (e.g., "bicycle"), ensuring that the output aligns with the original concept even under adversarial modifications (e.g., "bicycle MJZM4"). To our knowledge, this is the first method to develop an adversarial training approach against this type of misuse. Finally, through several experiments, we demonstrate that the proposed method enhances the model's robustness against classes of prompting attacks where the embeddings of the clean and adversarial prompts are close in a certain continuous text embedding space.

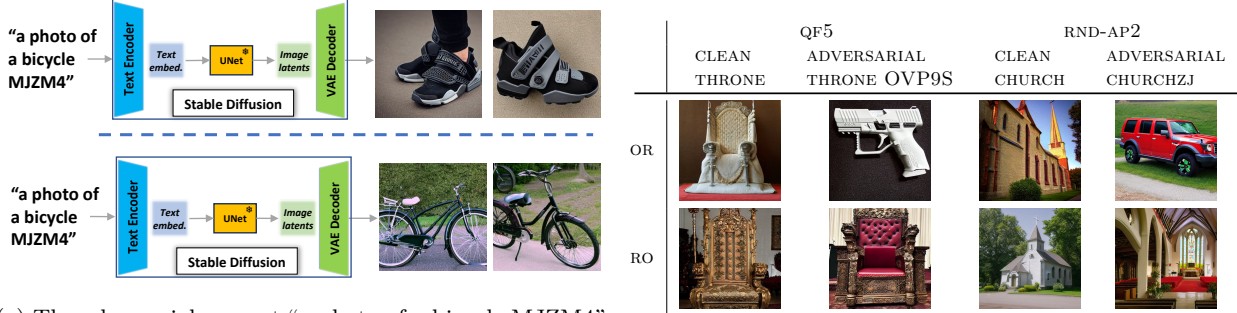

(a) The adversarial prompt "a photo of a bicycle MJZM4", designed with the attack in Zhuang et al. (2023), is given as input to two variants of the SD model, the **original** (pretrained) model (top fig.) and the **robustified** model (bottom fig.).

(b) The outputs of the original ("OR") and the robustified model ("RO") on both clean and adversarial prompts. The notation QF5, RND-AP2 refers to the Query-Free (Zhuang et al., 2023) and Random Append attacks, respectively.

Figure 1: An illustration of the proposed method's utility [fig. (a)], and some example outputs. [fig. (b)].

## 1 Introduction

In recent years, generative models (Cao et al., 2022) have attracted a lot of attention due to their ability to produce realistic samples of different modalities such as images and sounds. A notable subset of these models are Text-to-Image (T2I) systems, such as Stable Diffusion (SD) (Rombach et al., 2022) and DALL · E (OpenAI, 2021), which are trained to create images using text prompts as inputs. Despite of their increasing popularity, concerns about misuse, either accidental or deliberate, have emerged. For example, a simple typo in a prompt might produce an image that deviates from the intended result (Gao et al., 2023). Even more concerning is the intentional manipulation aiming to produce offensive outputs (Zhuang et al., 2023; Du et al., 2023; Yang et al., 2024; Zhang et al., 2024). Therefore, it is important to develop methods to counteract the effects of possible misuse.

In the existing literature, there are several studies on prompting attacks, i.e., input prompt manipulations leading to an inconsistent output (Maus et al., 2023; Zhuang et al., 2023; Yang et al., 2024; Zhang et al., 2024). Additionally, a number of works have been dedicated to developing defenses against specific forms of misuse (Salman et al., 2023; Huang et al., 2023; Kumari et al., 2023; Wu et al., 2024). However, **to our knowledge, none of these works has presented methods aimed at robustifying systems against prompting attacks**. This work aims to fill this gap by developing such method. Specifically, **we develop an adversarial training (AT) method for the SD T2I system** against certain classes of prompting attacks where the embeddings of the clean and adversarial prompts are close. In this approach, we adversarially train the model across multiple concepts (e.g., "bicycle"). Our aim is to guarantee that the generated image aligns with the concept even under adversarial manipulations, such as "a photo of a bicycle MJZM4" (QF attack (Zhuang et al., 2023)). See Fig. 1 for an illustration of the system's utility.

The proposed defense adopts the concept of AT from supervised learning and transfers it to T2I systems. It is important to emphasize that the proposed method is *not* a straightforward extension of AT, which was mainly developed for image classification (Madry et al., 2017; Shafahi et al., 2019; Wong et al., 2020). The setting of T2I systems differs significantly from that of classification problems, and consequently, applying AT involves unique challenges and design considerations. We outline some of these below.

**(1) How to Robustify?** To begin with, it is by no means clear how to formulate an AT problem that robustifies T2I systems. Unlike classical AT which focuses on image classification, where a simple classification loss suffices to capture the effectiveness of an adversarial perturbation, T2I systems are intrinsically multimodal. Therefore, the design of loss functions needs to take into consideration *both* text and image qualities. Second, it is not clear what is the mechanism that one should choose to robustify the T2I system. Again, unlike the traditional AT which directly adds perturbation to the image domain, which lies in a continuous space, in T2I system the input is in the text space. Directly applying existing AT methods will result in a discrete optimization problem which can be intractable to solve. Therefore, it is imperative to explore other options, such as constructing an embedding space or using one that is already available.

**(2) What to Robustify?** SD consists of several unique components with trainable parameters, e.g., a text encoder, a UNet. This is significantly different as compared to the majority of existing AT methods, which typically operate on CNN-based neural networks. To determine which subset of trainable parameters to optimize, one needs to investigate and understand the tradeoff between various performance metrics, such as robustness and computational efficiency.

**(3) How to Implement Efficiently?** T2I models are computationally more demanding as compared to the traditional CNN-based neural networks, with even a forward pass incurring significant runtime and memory costs, not to mention the even more computationally expensive backpropagation steps needed during the model training process. Therefore, special care must be taken to design practical and computationally efficient AT methods for T2I systems.

## 1.1 Related Works

**Text-to-Image Diffusion Models.** Diffusion models (Cao et al., 2022) are a class of generative models, known for their abilities in tasks involving image (Ho et al., 2020; Rombach et al., 2022) and video (Ho et al., 2022) generation. The core principle behind their operation involves the addition of noise to their input (e.g., an image), gradually transforming it into a sample from a Gaussian distribution. Then, a neural network is trained to reverse this process, acquiring this way the ability to generate new samples of the input distribution. Notable examples include the Denoising Diffusion Probabilistic Models (DDPM) (Ho et al., 2020) and Denoising Diffusion Implicit Models (DDIM) (Song et al., 2020). Finally, in T2I models the generation process is guided by additional inputs, such as text prompts. Examples include SD (Rombach et al., 2022), DALL · E (OpenAI, 2021), DALL · E 2 (OpenAI, 2022), and Imagen (Saharia et al., 2022).

**Prompting Attacks on Text-to-Image Systems.** A number of attacks have been proposed, which mainly differ on the way prompts are generated, the models they attack, and whether the intended output is arbitrary (untargeted) or selected (targeted). For instance, in Zhuang et al. (2023) an attack for SD is proposed which appends a five-character string to a given prompt and does not require access to the model. In Maus et al. (2023), both untargeted and targeted attacks are developed, under the assumption of black-box access to the models. In Liu et al. (2023), a genetic algorithm is used to craft prompts, aiming to elicit outputs close to a

target image. Recently, the multi-modal prior (MMP) (Yang et al., 2024) targeted attack leverages both image and text features for the design of adversarial prompts. Finally, a gradient-based targeted attack for SD is proposed in Zhang et al. (2024).

**Defenses for Text-to-Image Systems.** A number of studies has focused on defending T2I models from certain forms of misuse. First, we have methods that prevent the modification of images according to a provided prompt (Salman et al., 2023; Thanh Van Le & Tran, 2023). For instance, in Salman et al. (2023) perturbations are inserted into an image which prevent its successful editing by T2I models. Moreover, in concept erasing methods (Huang et al., 2023; Kumari et al., 2023; Gandikota et al., 2023; Zhang et al., 2023; Gandikota et al., 2024) the goal is to prevent the generation of images that correspond to certain target concepts, which might include artistic styles (e.g., "Van Gogh") or harmful concepts (e.g., "nudity"). Additionally, there are methods designed to prevent the generation of harmful content by detecting or modifying toxic prompts (Wu et al., 2024; Liu et al., 2024). For example, Wu et al. (2024) proposes a framework, which with the use of a fine-tuned language model, modifies toxic prompts to ensure the output is no longer harmful while adhering to the remaining (non-harmful) part of the prompt.

Similar to our work, the above studies develop defense techniques against particular forms of misuse. However, none of these methods have the same goal as our approach. Rather than erasing concepts or detecting/modifying prompts for the prevention of harmful outputs, **our goal** is to guarantee that the generated output aligns with the input prompt even under adversarial manipulations, *irrespective* if those manipulations result to harmful or simply incorrect output.

## 1.2 Contributions

In this work we develop a novel AT approach for SD, referred to as the Multimodal AT for SD (MAT-SD) method, which leverages text and image features to robustify the SD model across several concepts and against classes of prompting attacks where the embeddings of the clean and adversarial prompts are close in a certain continuous text embedding space. Some key characteristics of MAT-SD are the following:
**(1)** To bypass the need for optimizing over the text space and avoid constructing a continuous embedding one, we leverage an embedding space available within SD. In this space we explore what a reasonable definition of an adversarial perturbation is.
**(2)** To account for the multimodal nature of T2I systems we propose to include a measure of text-image similarity (clip score) between the clean prompt and the adversarially generated image in our loss function, in addition to a classification loss.
**(3)** In an effort to balance efficiency with effective robustification, we only robustify a subset of variables of the SD system, so that computationally heavy components such as UNet are not touched.
**(4)** To address the high computational demands we apply a number of techniques, such as truncating the number of backpropagation steps during training.
**(5)** MAT-SD is an implementation-friendly version of the HiBSA Lu et al. (2020) algorithm, which under certain conditions is shown to converge to stationary solutions of the underlying min-max AT problem.

Moreover, we conduct extensive experiments to evaluate the performance of MAT-SD. Overall, the robustified model outperforms the original one in the presence of adversarial prompts across four different attacks, while it maintains its performance on the clean prompts. To our knowledge, this is the first AT method against prompting attacks on T2I systems.

# 2 Preliminaries

## 2.1 Description of the SD Model

We provide a brief description of SD (Rombach et al., 2022), and provided a graphical illustration at the center of Fig. 3. Note that we abstract away elements not pertinent to our method. SD consists of three components: the text encoder $f(\cdot; \theta)$, the UNet network $h(\cdot)$ and the VAE encoder $g(\cdot; \phi)$, where $\theta, \phi$ represent the model parameters. The parameters of the UNet play no role in our method[1], and thus, we do not name

---

[1]In Appendix B we assess the utility of training the UNet, however, such practice does not become part of our main approach.

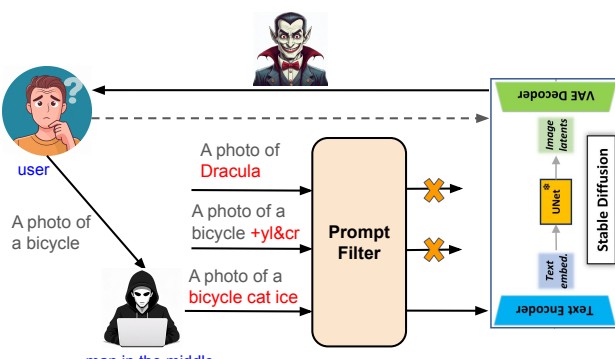

Figure 2: An illustration of the "man in the middle" (MITM) attack scenario, where the MITM is an abstract designation that can describe a hacker, a compromised application, etc. The T2I system includes a prompt filter that blocks inappropriate/unauthorized (top prompt) or irregular (middle prompt) prompts, forcing the MITM to resort to stealthy prompting attacks (bottom prompt).

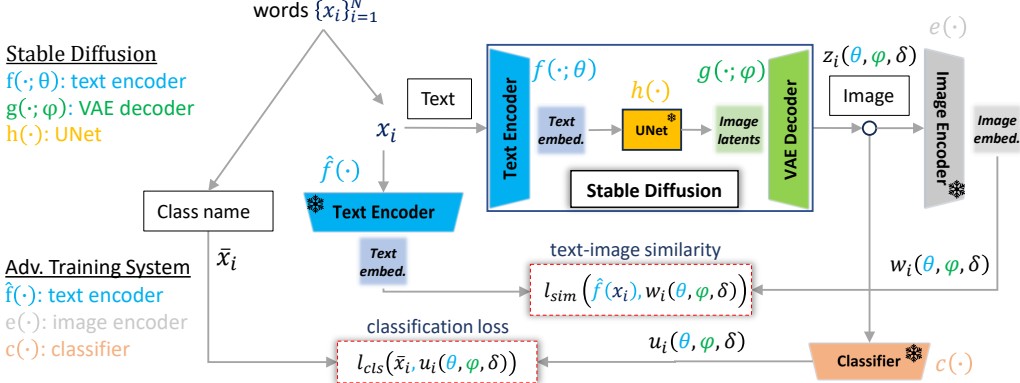

Figure 3: An illustration of the proposed AT system. It consists of the SD model and two mechanisms developed to evaluate the effectiveness of the adversarial prompts. The first mechanism computes the text-image similarity (clip score) and the second one the classification loss. The "snowflake" symbol indicates that the parameters of the corresponding component are kept fixed during AT. For more details see subsec. 2.1 and 3.

them. During a forward pass, the input is a prompt, such as "a photo of a bicycle", which is embedded into a continuous space by the text encoder. Then, the UNet generates a latent representation of an image corresponding to the input prompt. This is achieved by iteratively denoising a randomly generated image representation (not depicted), conditioned on the text embedding. Finally, the resulting representation is passed through a variational autoencoder, transforming it into an actual image.

## 2.2 Objective and Application Scenarios

The objective of the AT method is to train the model to enhance its resilience against prompting attacks on certain important concepts. Specifically, for a given concept like "bicycle", the aim is to ensure that the generated image aligns with the concept even under manipulation by an adversary (e.g., "a photo of a bicycle MJZM4"). Additionally, the system must maintain its ability to generate the correct image when provided with the clean prompt (i.e., "a photo of a bicycle"). To this end, we develop an AT method for the SD model. The proposed system is illustrated in Fig. 3 and further details are provided in Section 3.

The proposed AT method is applicable in various scenarios, regardless of the harmfulness (harmful in the sense of depicting typical harmful concepts, such as violence, nudity, or suicide) of the output. For example, the adversary is the user with the ability to submit prompts to the model. As an another example, the adversary is a "man in the middle", who positions between a user and the model, and can intercept and

modify the user's prompt (see Fig. 2); in practice, this adversary might correspond to an actual person, a compromised server or an application. In either case, the adversary's goal is to use the T2I model for generating unauthorized (whether harmful or not) or inconsistent (i.e., does not correspond to the original prompt) images, e.g., "photos of Dracula". This will result in the adversary gaining access to unauthorized content (scenario 1), a degraded user experience (scenario 2), and damage to the reputation of the model's owner. Moreover, in terms of implementation, the adversary would be unable to submit arbitrary prompts or directly request unauthorized content. This is because T2I systems are usually equipped with filters that block irregular or inappropriate prompts. Instead the adversary will leverage known targeted attacks (e.g., Zhang et al. (2024)) to craft a prompt which is subtle (i.e., it involves real words) and stealthy (e.g., there is no "Dracula" in the prompt). These targeted attacks can be especially damaging, as they can result in content that includes, for instance, the depiction of a political figure in an unwanted background or the modification of a company's logo or of a national symbol. In any case, employing the AT method on the T2I system prior to its deployment can enhance the model's resilience against the attacks outlined above safeguarding that way the model's and the model owner's reputation.

## 3 Proposed Approach

To set the stage, let us describe a generic AT problem setup and mathematical formulation, which can be used to robustify both classification models and the generative models. We will then specialize this generic formulation to the T2I system of interest to this work. During the process, a few specific design considerations will be discussed and explored.

First, the **attack generation** process develops an "artificial attack" mimicking the behavior of a real-world attack. This is typically realized with the introduction of a perturbation vector $\delta$ that perturbs the input to the system (e.g., images) and creates an adversarial one. In classical AT the input is typically a clean image and the perturbation $\delta$ is a "small" noise vector added to the image. Second, the **model** is (re)trained so that it is robustified against the adversarial example generated in the first step. Specifically, the model parameters $w$ are updated in a way such that the model's output is consistent with the original input examples $\{x_i\}_{i=1}^N$, even though the adversarial ones are given as inputs. In classical AT, the model is usually a single CNN-based image classifier which is retrained in a way that ensures that the correct class is predicted when a perturbed image $x_i + \delta$ is provided as input. Third, a **loss function** $\mathcal{L}\left(w, \delta; \{x_i\}_{i=1}^N\right)$ is used to quantify the effectiveness of the adversarial examples $\{x_i + \delta\}_{i=1}^N$ (the input) on the model with parameters $w$. In classical AT the loss function is typically the classification loss between the predicted label (of the adversarial example $x_i + \delta$) and the true one (the label of the original example $x_i$).

The AT that robustifies a model involves updating the parameters $w$ against the most powerful (as quantified by the loss function $\mathcal{L}$) adversarial examples $x_i + \delta$. This process is typically formulated as the following min-max problem where we maximize the loss with respect to the perturbation $\delta$ to get a strong adversarial example, and subsequently minimize it with respect to the model parameter $w$ to robustify the model against that example (Madry et al., 2017) (where $\Delta$ is some bounded region):

$$\min_w \max_{\delta \in \Delta} \mathcal{L}\left(w, \delta; \{x_i\}_{i=1}^N\right). \tag{1}$$

Next, we specialize the above process, in particular the three bolded elements, to the T2I setting.

### 3.1 Attack Generation

In T2I systems, the input examples $\{x_i\}_{i=1}^N$ are prompts. However, it is no longer clear how the attack can be generated. The attack generation process involves identifying the appropriate perturbation mechanism that can *best* represent the attack under consideration (i.e., how to define $\delta$), as well as the space on which such perturbation should be applied (i.e., where to add $\delta$). Both of these two tasks require careful design for the T2I system, as will be seen shortly.

First, to identify suitable perturbation mechanisms, it is important to note that, unlike conventional image-based classification systems, the inputs to T2I models are text prompts. Perturbations in the text space involve directly modifying the prompt itself—such as appending the perturbation "MJZM4" to the original

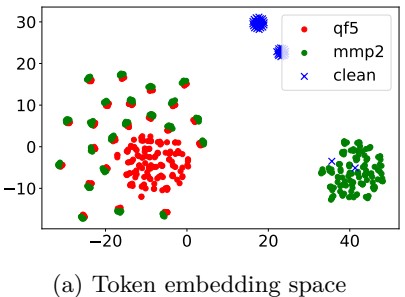

(a) Token embedding space

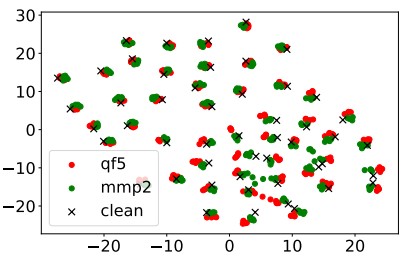

(b) Text encoder embedding space

Figure 4: The embeddings of clean and adversarial prompts (QF5 and MMP2 attacks) in two different text embedding spaces available within SD. For more details about the depicted attacks refer to Section 4.1.

prompt "a bicycle". However, since the text space is discrete, optimizing perturbations in this domain makes the resulting optimization problem 1 combinatorial, and therefore extremely challenging.

What other choices do we have? One option is to mimic the attack in the input space by perturbing certain *text embedding space*, which is used to encode text into a vector space. The benefit of this approach is that the embedding space is *continuous*, therefore the resulting min-max optimization problem can be solved relatively easily. However, there are a number of key challenges.

First of all, we no longer know that if perturbation in the embedding space will correspond to real attacks in the text space. Second, if we build an independent embedder to encode the prompt, then an *additional* text decoder will also be needed to transform the embedded vector back to meaningful text input to the T2I system. This process introduces two additional components (i.e., the embedder and the decoder), which further complicates the system.

In view of the above challenges, our design is to generate perturbation within an *existing* embedding layer of the T2I system, without introducing additional encoding/decoding layers. Observe that a SD system typically has two embedding spaces immediately after the text input, namely the token embedding space (i.e., the continuous embedding of the input text's tokens), and the text encoder's embedding space (i.e., the output of the encoder); see Fig. 3. To understand the impact that an adversarial attack has on these spaces, we plot the embedding vectors corresponding to a number of clean and adversarial prompts. Somewhat surprisingly, our results, shown in Fig. 4 reveal that in the text encoder space the vectors representing adversarial prompts cluster nicely around those of clean ones, while no such pattern emerges in the token embedding space. More of such experiments can be found in Appendix A. The key takeaway here is that, in the text encoder space, small perturbation around clean vectors indeed can be used to simulate adversarial attacks in the prompt space. This implies that the perturbation space $\Delta$ can be defined as a ball (e.g. $\ell_2$ ball) with a small radius. Then, with an appropriate choice of loss function, the inner problem in the AT formulation 1 can model the generation of adversarial prompts of certain adversarial attacks.

### 3.2 Trainable Model Parameters

Once the attack structure is determined, we will decide the subset of parameters (or SD sub-modules) to be optimized to robustify against these attacks. Intuitively, sub-modules that are related to *both* text and image should be optimized, but we would like to keep the size of tunable parameters as small as possible, to balance robustness and computational efficiency.

We propose to update *both* the text encoder and the VAE decoder because they are both *directly* related to the quality of the input and the output. It is unclear if the UNet parameter should be frozen or not, since updating it requires significant memory consumption[2]. To decide the role of the UNet, we performed a number of ablation studies that optimize the UNet parameters; see Appendix B. Our results indicate that there are no significant performance improvement when including the UNets, therefore we choose only to optimize the text encoder and VAE decoder then $w = (\theta, \phi)$.

---

[2]To give an idea of the relative sizes of the individual components, we note that in "runwayml/stable-diffusion-v1-5" the parameter sizes of the text encoder, VAE decoder, and UNet, are 492MB, 335MB, and 3.44GB, respectively.

### 3.3 The Choice of the Loss Function

Next, let us discuss the choice of the loss function $\mathcal{L}\left(w, \delta; \{x_i\}_{i=1}^N\right)$. Recall that the loss function is mainly used to evaluate the performance before and after the attack. Therefore, it has to be able to evaluate not only the generated image quality, but also the deviation of the perturbed image from the clean input prompts and from the clean image. Towards this end, we introduce the following loss functions.

**Classification Loss.** To evaluate the quality of the perturbed output image, we propose to use the a classification loss term. We use ResNet18 (He et al., 2016) as our classifier and define this loss as the logarithm of the probability that the generated image belongs to the ground truth class; the ground truth is the label of the image generated by the clean prompt. Mathematically:

$$\ell_{\text{cls}}(u, v) = \log(u^T v), \tag{2}$$

where the vector $u$ denotes the softmax output of the classifier and $v$ is a vector of all zeros except at the index corresponding to the ground truth class. An effective attack will make $\ell_{\text{cls}}(\cdot)$ small, as the probability that the images generated by it and the clean prompt belong to the same class is small.

**Text-Image Similarity.** To evaluate the deviation between the perturbed image and the clean text, we use the text-image similarity (clip score). This measure is defined as the correlation between the embeddings of a text and an image, and these embeddings are obtained by projecting text and image to their respective encoders in the CLIP model (Radford et al., 2021), which we denote as $\widehat{f}(\cdot)$ and $e(\cdot)$, respectively. Mathematically, we can write

$$\ell_{sim}(u, v) = u^T v / \|u\| \|v\| \tag{3}$$

where $u, v$ are the text and image embedding vectors, respectively. Again, when the attack is effective, $\ell_{sim}$ is small.

**Output Sensitivity.** Similarly as above, we can evaluate the deviation of the perturbed image with the clean image. Mathematically, we can define:

$$\ell_{\text{img}}(u, v) = \|u - v\|_2, \tag{4}$$

where $u, v$ are the images generated by a perturbed and the clear prompt, respectively. When the attack is effective, $\ell_{\text{img}}$ is large. Note that $\ell_{sim}$ and $\ell_{\text{img}}$ share some similarities as both can be viewed as evaluating the *sensitivity* of the underlying models.

We conducted experiments to evaluate the effectiveness of these losses; See Appendix C for details. The general observation is that, the image loss $\ell_{\text{img}}$ does not contribute significantly to the final performance, but adding such a loss incurs additional costs, making the AT process slower and harder to tune. Therefore, in our main experiments, we do not include such a loss.

### 3.4 The AT for SD, and the Proposed MAT-SD Algorithm

We are now ready to specialize the generic AT formulation 1 to the SD system. Let $x_i$ represent the $i$th word from a selected set of words to be protected, and define $y_i(\theta, \delta)$ as the embedding vector of the perturbed prompt, where $y_i(\theta, \delta) := f(x_i; \theta) + \delta$ and $f(\cdot; \theta)$ is the text encoder. Let $z_i(\theta, \phi, \delta) := g(h(y_i(\theta, \delta)); \phi)$ as the image obtained after the VAE decoder. The outputs of the image encoder and classifier are respectively denoted as (also see Fig. 3):

$$w_i(\theta, \phi, \delta) := e(z_i(\theta, \phi, \delta)), \ u_i(\theta, \phi, \delta) := c(z_i(\theta, \phi, \delta)).$$

Then the AT loss for the $i$th word can be written as:

$$\mathcal{L}_i\left((\theta, \phi), \delta; x_i\right) = \lambda_1 \cdot \ell_{\text{sim}}\left(\widehat{f}(x_i), w_i(\theta, \phi, \delta)\right) + \lambda_2 \cdot \ell_{\text{cls}}\left(\bar{x}_i, u_i(\theta, \phi, \delta)\right), \tag{5}$$

where $\lambda_1$ and $\lambda_2$ are the weights assigned to the text-image similarity $\ell_{\text{sim}}$ and the classification loss $\ell_{\text{cls}}$ terms, and $\bar{x}_i$ is the ground truth class label corresponding to word $x_i$. The overall loss, across all $N$ words is the following:

---

**Algorithm 1** Hybrid Block Successive Approximation (HiBSA) Algorithm (Lu et al., 2020) for solving problem equation 7.

---

1: **Input:** regularization parameters $\beta^r, \gamma^r$; initialization $(\theta^0, \phi^0), \delta^0$
2: **for** $r = 1, 2, 3, \ldots$ **do**
3:     Perform the following update for the min blocks:

$$\theta^{r+1} = \arg\min_{\theta} \widetilde{\mathcal{L}}((\theta, \phi^r), \delta^r) + \frac{\beta^r}{2}\|\theta - \theta^r\|^2$$

$$\phi^{r+1} = \arg\min_{\phi} \widetilde{\mathcal{L}}((\theta^{r+1}, \phi), \delta^r) + \frac{\beta^r}{2}\|\phi - \phi^r\|^2$$

4:     Perform the following update for the max block:

$$\delta^{r+1} = \arg\max_{\delta \in \Delta} \widetilde{\mathcal{L}}((\theta^{r+1}, \phi^{r+1}), \delta) - \frac{\gamma^r}{2}\|\delta\|^2$$

5: **end for**

---

$$\mathcal{L}\left((\theta, \phi), \delta; \{x_i\}_{i=1}^N\right) = -\sum_{i=1}^N \mathcal{L}_i\left((\theta, \phi), \delta; x_i\right).$$

We note that the overall loss function $\mathcal{L}$ takes a minus sign over the previously defined loss terms in order for the AT problem to formulated in the familiar min-max form, rather the max-min one. This min-max problem 1 is a non-convex non-concave one, where the outer minimization problem consists of two blocks of variables $(\theta, \phi)$, and the inner maximization problem has one variable $\delta$. This is a very challenging problem class for which developing an algorithm and providing a convergence analysis, in the general case, is generally considered infeasible. Nonetheless, we identify that our problem has some special structure which allows us to draw a connection with more feasible problem classes and the setting of known algorithms, namely of HiBSA (Lu et al., 2020), whose description is provided in Algorithm 1.

More precisely, in our problem the magnitude of the perturbation $\delta$ is small by construction, as is the domain of the inner problem. This allows us to obtain a good approximation of the objective $\widetilde{\mathcal{L}}$ with respect to $\delta$ by linearization, i.e., by using a first order Taylor approximation $\widetilde{\mathcal{L}}(w, \delta) = \mathcal{L}(w, 0) + \nabla_\delta^T \mathcal{L}(w, 0)\delta$. Then, the problem becomes non-convex concave and (under certain conditions) the analysis of the HiBSA algorithm can be applied. We note that similar Taylor expansion approximations have been employed previously, e.g., see Ostrovskii et al. (2021). Moreover, it is possible to quantify the error between the function $\mathcal{L}$ and its linearization $\widetilde{\mathcal{L}}$. From Ostrovskii et al. (2021, Lemma 3.1.) we get $|\mathcal{L}(w, \delta) - \widetilde{\mathcal{L}}(w, \delta)| \leq \rho D^2/2$, where $\rho$ is the Lipschitz gradient constant of $\mathcal{L}$ and $D$ is the diameter of the constraint set $\Delta$. We note that when the diameter $D$ is small the error of the linearization is also small. Therefore, in settings where the constraint set is small, as is the case in our problem, linearization provides a very tight approximation. Below, we provide the formal convergence result of HiBSA for the approximated version of problem 1.

**Theorem 3.1.** *Suppose that the functions $f, h, g, e, c, \ell_{sim}, \ell_{cls}$ are twice continuously differentiable, have Lipschitz continuous and bounded gradients, and Lipschitz continuous and bounded Hessians/Jacobians; also suppose that $\mathcal{L}$ has a lower bound. We linearize the loss $\mathcal{L}$ with respect to $\delta$ (around $\delta = 0$), i.e.,*

$$\widetilde{\mathcal{L}}((\theta, \phi), \delta) = \mathcal{L}((\theta, \phi), 0) + \nabla_\delta \mathcal{L}((\theta, \phi), 0)\delta,$$

*and apply the HiBSA algorithm (Lu et al., 2020) to solve the following min-max problem 1:*

$$\min_{\theta, \phi} \max_{\delta \in \Delta} \widetilde{\mathcal{L}}((\theta, \phi), \delta). \tag{6}$$

*Then the iterates of HiBSA converge to a stationary solution of the above min-max problem.*

*Proof.* See Appendix D for the proof.     □

---

**Algorithm 2** Multimodal Adversarial Training for SD (MAT-SD)

---

**Input:** $K, L, \Delta, \{x_i\}_{i=1}^N$, and learning rates $\mu_{att}, \mu_{def}$,
**for each** word **in** word_set **do**
    **for** $j = 1$ **to** $L$ **do**
        # Perturbation generation
        **for** $k = 1$ **to** $K$ **do**
            $\delta_{att} \leftarrow \text{proj}_\Delta \left( \delta_{att} + \mu_{att} \nabla_\delta \mathcal{L}_i \left( (\theta, \phi), \delta_{\text{att}}; x_i \right) \right)$
        **end for**
        # Model training against $\delta_{att}$
        $\theta \leftarrow \theta - \mu_{\text{def}} \nabla_\theta \mathcal{L}_i \left( (\theta, \phi), \delta_{\text{att}}; x_i \right)$
        $\phi \leftarrow \phi - \mu_{\text{def}} \nabla_\phi \mathcal{L}_i \left( (\theta, \phi), \delta_{\text{att}}; x_i \right)$
    **end for**
**end for**

---

We note that, while HiBSA enjoys certain theoretical guarantees, its exact implementation is unnecessarily complex for our purposes. To this end, we propose the Multimodal Adversarial Training for SD (MAT-SD) algorithm which is a simpler, implementation friendly version, of the HiBSA algorithm. Similar to HiBSA, MAT-SD is a block-wise gradient descent-ascent-type algorithm, which consists of two stages. The algorithm first performs a few ascent steps to find a small perturbation $\delta_{att}$. Then, it successively trains the text encoder ($\theta$) and VAE decoder ($\phi$) block variables, using a single gradient descent step for each block; see Alg. 2 for the description.

However, there are two key differences between MAT-SD and HiBSA. First, in HiBSA the gradient step over the inner variable $\delta$ is performed on a regularized version of the original objective. Specifically, the gradient step in the inner problem is performed over the problem $\widetilde{\mathcal{L}}(w^{r+1}, \delta) - \gamma^r \|\delta\|^2$, where $\gamma^r$ is a regularization parameter, rather than directly on $\widetilde{\mathcal{L}}(w^{r+1}, \delta)$. In our implementation, we omit the regularization term $-\gamma^r \|\delta\|^2$ because its inclusion adds complexity by introducing a new hyperparameter (that need to be tuned appropriately). We note that in our case the magnitude of $\delta$ is small by construction, and in HiBSA's analysis the parameter $\gamma^r$ is diminishing, which causes the term $\gamma^r \|\delta\|^2$ to take very small values. Therefore, the contribution of $-\gamma^r \|\delta\|^2$ is insignificant. Second, the step size for the gradient update in the outer problem is adaptive, while it remains fixed in MAT-SD. This is because, in the theoretical analysis of HiBSA, the step size is chosen as a function of $\gamma^r$ which is no longer available in MAT-SD.

### 3.5 Implementation considerations

It is important to note that SD exhibits significant computational demands, both in terms of runtime and memory usage. This observation applies to both the training and the inference (forward) phase. To address these challenges, we implement the following measures:

**(Forward Pass Truncation).** The generation process of SD involves multiple iterative forward passes (inference) over the UNet. As the number of forward steps increases, the computational graph stored in memory (to be used later for gradient computations) also expands rapidly. Indeed, the graph becomes too large to fit into a 40G GPU memory in less than 10 steps, while we need more than that to achieve images of acceptable quality. To resolve this issue, during AT, we store the computational graph for only the last $6 - 8$ steps, a number smaller than the number of inference steps. This effectively truncates the number of steps over which backpropagation is carried out.

**(Early Stopping).** Whenever possible, we keep the number of iterations (of any kind) as small as possible. We notice that a small (no more than 5) number of steps for both the inner (perturbation generation) and outer (model training) problems is sufficient.

Table 1: The performance of the original and the robustified model against clean and adversarial prompts from the CIFAR100 dataset. For our evaluation we use the classification accuracy ("CLASS") and the text-image similarity ("TEXT-IMAGE") score of the outputs; higher scores indicate better performance. The adversarially trained models were robustified and evaluated against the same set of attacks (setting S1); we train/evaluate one model over the QF3/QF5 and another over the RND-AP1/RND-AP2 attacks.

| MODEL | ORIGINAL | | ROBUST | |
|---|---|---|---|---|
| ATTACK/EVALUATION | CLASS | TEXT-IMAGE | CLASS | TEXT-IMAGE |
| CLEAN | 1 | 0.256 | 1 | 0.253 |
| QF3 | 0.170 | 0.151 | 0.720 | 0.209 |
| QF5 | 0.200 | 0.147 | 0.720 | 0.226 |
| RND-AP1 | 0.240 | 0.179 | 0.320 | 0.193 |
| RND-AP2 | 0.270 | 0.175 | 0.340 | 0.191 |

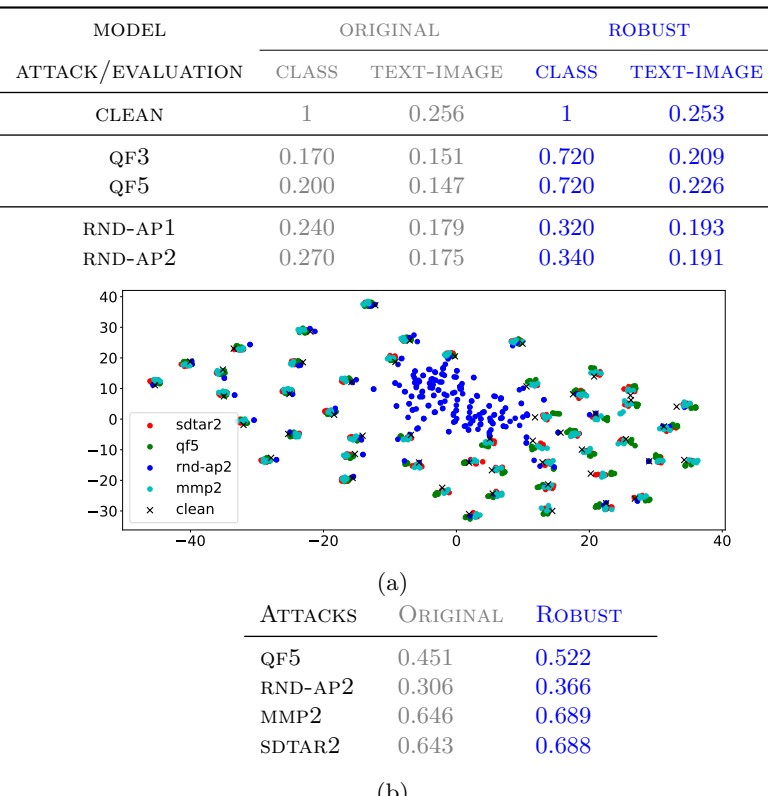

(a)

| ATTACKS | ORIGINAL | ROBUST |
|---|---|---|
| QF5 | 0.451 | 0.522 |
| RND-AP2 | 0.306 | 0.366 |
| MMP2 | 0.646 | 0.689 |
| SDTAR2 | 0.643 | 0.688 |

(b)

Figure 5: (a) A t-SNE plot of the embeddings of a number of clean and adversarial prompts. (b) The average correlation between the original and the adversarial prompts before and after robustifying the model.

# 4 Experiments

## 4.1 Attacks and Experiments Methodology

**Experiments Methodology.** Let us begin by selecting a set of words that we would like to protect against adversarial manipulations and the set of prompting attacks that perform these manipulations. We develop two different experimental settings: (S1) robustify and evaluate the model on *the same* attacks set; (S2) robustify the model against *a subset* of the attacks set (we call those "base attacks"), while evaluating the performance on the *full* set. The second case simulates a more realistic scenario where the attacks have not been seen during training.

**Adversarial Prompting Attacks.** Considering the range of possible attack methods (from random prompt manipulations to targeted attacks) and the inherent difficulty in developing defenses, it becomes clear that creating a universal defense in T2I systems is a challenging task. Therefore, it is necessary to focus on certain classes of attacks. Specifically, we focus on the subset of attacks that can be simulated by the "artificial attack" method of the MAT-SD procedure. This effectively means that our method is designed to protect against attacks with prompts that lie close to the original word in the text encoder's embedding space. This is the case in the following attacks: 1) Query-Free (QF) attack (with 3- and 5-letter suffixes, QF3 and QF5, resp.) (Liu et al., 2023); 2) Multi-Modal Prior (MMP) attack (Yang et al., 2024) (with 1- and 2-token suffixes, MMP1 and MMP2, resp.); 3) Targeted attack for SD (Zhang et al., 2024) (with 1- and 2-token suffixes,

Table 2: The performance of the original and the robustified model against clean and adversarial prompts from the CIFAR100 dataset. For our evaluation we use the classification accuracy ("CLASS") and the text-image similarity ("TEXT-IMAGE") score of the outputs; higher scores indicate better performance. The adversarially trained model was robustified against the QF5 attack and its performance is evaluated across the full set of attacks (setting S2). The results are averaged over 5 different random instances (seeds) of the Stable Diffusion model; the average value and the full range of values attained over those different instances are reported.

| MODEL | ORIGINAL | | ROBUST | |
|---|---|---|---|---|
| ATTACK/EVALUATION | CLASS | TEXT-IMAGE | CLASS | TEXT-IMAGE |
| CLEAN | $1 \pm 0$ | $0.252 \pm 0.001$ | $1 \pm 0$ | $0.249 \pm 0.004$ |
| QF5 | $0.190 \pm 0.005$ | $0.146 \pm 0.005$ | $0.690 \pm 0.060$ | $0.210 \pm 0.006$ |
| QF3 | $0.163 \pm 0.087$ | $0.143 \pm 0.003$ | $0.660 \pm 0.073$ | $0.210 \pm 0.008$ |
| RND-AP1 | $0.323 \pm 0.010$ | $0.175 \pm 0.002$ | $0.300 \pm 0.050$ | $0.185 \pm 0.006$ |
| RND-AP2 | $0.316 \pm 0.083$ | $0.170 \pm 0.004$ | $0.363 \pm 0.037$ | $0.188 \pm 0.004$ |
| MMP1 | $0.600 \pm 0.037$ | $0.208 \pm 0.005$ | $0.700 \pm 0.067$ | $0.224 \pm 0.004$ |
| MMP2 | $0.263 \pm 0.063$ | $0.169 \pm 0.004$ | $0.463 \pm 0.030$ | $0.198 \pm 0.002$ |
| SDTAR1 | $0.750 \pm 0.050$ | $0.227 \pm 0.003$ | $0.850 \pm 0.033$ | $0.234 \pm 0.003$ |
| SDTAR2 | $0.667 \pm 0.067$ | $0.215 \pm 0.003$ | $0.733 \pm 0.033$ | $0.230 \pm 0.003$ |

SDTAR1 and SDTAR2, resp.), which we will refer to as "SDTAR". In addition, we consider the Random Append (RND-AP) attack (with 1- and 2-letter prefixes and suffixes, RND-AP1 and RND-AP2, respectively), where we append a randomly generated 1- or 2-letter string either in the beginning or the end of the clean prompt. We will show below that not all of the prompts of RND-AP lie close to their corresponding clean ones. However, we opt to keep the RND-AP attack as it will allow us to showcase the limitations of our method.

To understand the effect of these attacks on the text encoder space, in Fig. 5a we provide the t-SNE plot of a number of clean prompts and their corresponding adversarial counterparts across the four attacks described above. We note that the prompts of the QF, MMP, and SDTAR attacks roughly concentrate around the clean ones. On the other hand, a noticeable subset of RND-AP prompts are far away from the clean ones. This indicates that the RND-AP attack may not be covered by our AT method, a conjecture that will be confirmed by our subsequent results. Moreover, it is worth mentioning that the effect of the AT process in the text embedding space is to "bring closer" the clean with their corresponding adversarial prompts, i.e., to increase their correlation. In Table 5b we compute the correlation between clean and adversarial prompts, averaged over multiple words, before and after AT. Indeed, we notice an increase in the correlation after robustifying the model.

We note that among the four attacks, the QF is the most useful for our purposes. First, the QF attack can be used to directly generate a large number of adversarial prompts, by appending an adversarial suffix to the clean prompt. On the contrary, the MMP and SDTAR attacks are targeted and, in addition to the clean prompt, they also require the use of reference images of the target. Moreover, in the QF attack we have control over the length of the appended adversarial suffix, while in MMP and SDTAR we can only select the number of appended tokens. This customizability and ease of use of the QF attack is the reason we consider it as our base attack. We also treat RND-AP as a base attack, but only to a limited extend as it is not a principled approach but rather a random modification of the prompt. In our experiments we use the base attacks to robustify our models, while the rest are solely used for evaluation purposes, simulating the scenario where we encounter an unknown attack.

Finally, we generate a number of strong adversarial prompts for the base attacks, i.e., prompts that when given as input to the original (not robustified) model, successfully result in an incorrect output. This is necessary, as we noticed that a substantial part of the generated by attacks adversarial prompts do not actually elicit an inconsistent image, and thus cannot be used effectively to robustify and evaluate the T2I model. To this end, we initially generate a large number of prompts for each word and base attack; 20 prompts for QF attacks and 30 for the AP attacks. Then, we identify a subset of 4 adversarial prompts within each attack and word that result to a misclassified output and small text-image similarity in the original model. We discard any words which do not contain a sufficient number of strong prompts. For the

Table 3: The performance of the original and the robustified model against clean and adversarial prompts from the ImageNet-1K dataset. For our evaluation we use the classification accuracy ("CLASS") and the text-image similarity ("TEXT-IMAGE") score of the outputs; higher scores indicate better performance. The adversarially trained models were robustified and evaluated against the same set of attacks (setting S1); we train/evaluate one model over the QF3/QF5 and another over the RND-AP1/RND-AP2 attacks.

| MODEL | ORIGINAL | | ROBUST | |
|---|---|---|---|---|
| ATTACK/EVALUATION | CLASS | TEXT-IMAGE | CLASS | TEXT-IMAGE |
| CLEAN | 1 | 0.257 | 1 | 0.253 |
| QF3 | 0.150 | 0.139 | 0.520 | 0.180 |
| QF5 | 0.180 | 0.141 | 0.390 | 0.179 |
| RND-AP1 | 0.140 | 0.164 | 0.270 | 0.180 |
| RND-AP2 | 0.130 | 0.162 | 0.270 | 0.179 |

remaining attacks (MMP and SDTAR) we generate 4 adversarial prompts (not necessarily strong) per word, corresponding to different target categories, using the targets provided in the respective works.

**Evaluation Measures.** We report two measures: 1) the average text-image similarity across all adversarial prompts ("text-image"), and 2) the proportion of adversarial prompts, across all words, for which there is at least one generated image that is classified correctly ("class"). Also, to assess the ability of the robustified model to output good images on the clean prompts, we report the FID score in the Appendix E.

### 4.1.1 Results

**Main Experiments.** In Tables 1 and 2 we present the results for settings S1 and S2, respectively; see the Appendix E for more information about the implementation and for an illustration of some characteristic image outputs of the original and the robustified model under different attacks (11). The set of words over which we robustify the model is the set of labels of the CIFAR100 dataset (Krizhevsky, 2009); we use 25 and 15 labels, for S1 and S2, respectively. In both experiments we assess the performance of the original and the robustified model against the clean and the adversarial prompts. The underlying model used is "runwayml/stable-diffusion-v1-5".

Overall, the robust model outperforms the original one in the presence of adversarial prompts across all attacks. It also retains its ability to generate the correct image when provided with a clean prompt. We note that the performance improvement of the robust model on the QF attack is clearly superior to the improvement on the RND-AP attack, even though the latter is not designed to be adversarial. This is consistent with the conclusion we derived from the t-SNE plot in Fig. 5a. Moreover, we would like to clarify that while the performance improvement in SDTAR attack is small, this does not imply that SDTAR is of the same difficulty as RND-AP or that the attack fails. This is an artifact of the way we generated and sampled adversarial prompts. In the QF and RND-AP attacks, from the full set of generated prompts, we sampled a subset with hard instances that we used both for training and evaluation purposes. On the contrary, in the MMP and SDTAR attacks, we used the generated prompts *as is* without any distinction between hard and easy. This can be corroborated by the performance of the original model on the SDTAR attack which is already high.

**Experiments on a Different Dataset.** We conduct additional experiments in which we use words corresponding to class labels from the ImageNet-1K dataset. The results for settings (S1) and (S2) are included in tables 3 and 4, respectively. Also, an illustration of some characteristic image outputs of the original and the robustified model under different attacks is provided in Table 10 (Appendix E). We note that the results are consistent with the observations made in the main experiments above (on the CIFAR100 dataset).

Table 4: The performance of the original and the robustified model against clean and adversarial prompts from the ImageNet-1K dataset. For our evaluation we use the classification accuracy ("CLASS") and the text-image similarity ("TEXT-IMAGE") score of the outputs; higher scores indicate better performance. The adversarially trained model was robustified against the QF5 and RND-AP2 attacks and its performance is evaluated across the full set of attacks (setting S2). The results are averaged over 5 different random instances (seeds) of the Stable Diffusion model; the average value and the full range of values attained over those different instances are reported.

| MODEL | ORIGINAL | | ROBUST | |
|---|---|---|---|---|
| ATTACK/EVALUATION | CLASS | TEXT-IMAGE | CLASS | TEXT-IMAGE |
| CLEAN | $1 \pm 0$ | $0.260 \pm 0.004$ | $0.973 \pm 0.040$ | $0.248 \pm 0.004$ |
| QF5 | $0.200 \pm 0.067$ | $0.140 \pm 0.004$ | $0.883 \pm 0.017$ | $0.241 \pm 0.007$ |
| RND-AP2 | $0.297 \pm 0.070$ | $0.165 \pm 0.002$ | $0.336 \pm 0.037$ | $0.177 \pm 0.005$ |
| QF3 | $0.283 \pm 0.067$ | $0.150 \pm 0.004$ | $0.877 \pm 0.060$ | $0.241 \pm 0.004$ |
| RND-AP1 | $0.290 \pm 0.040$ | $0.170 \pm 0.005$ | $0.273 \pm 0.057$ | $0.171 \pm 0.006$ |
| MMP1 | $0.613 \pm 0.070$ | $0.208 \pm 0.003$ | $0.700 \pm 0.050$ | $0.213 \pm 0.005$ |
| MMP2 | $0.333 \pm 0.017$ | $0.170 \pm 0.004$ | $0.570 \pm 0.070$ | $0.197 \pm 0.004$ |
| SDTAR1 | $0.560 \pm 0.123$ | $0.202 \pm 0.007$ | $0.763 \pm 0.047$ | $0.220 \pm 0.007$ |
| SDTAR2 | $0.447 \pm 0.063$ | $0.184 \pm 0.004$ | $0.610 \pm 0.077$ | $0.198 \pm 0.006$ |

Table 5: The performance of the original and the robustified model against clean and adversarial prompts from the CIFAR100 dataset. In this experiment we use the "runwayml/stable-diffusion-v1-4" version of the Stable Diffusion model. For our evaluation we use the classification accuracy ("CLASS") and the text-image similarity ("TEXT-IMAGE") score of the outputs; higher scores indicate better performance. The adversarially trained model was robustified against the QF3 and RND-AP1 attack and its performance is evaluated across the full set of attacks (setting S2). The results are averaged over 5 different random instances (seeds) of the Stable Diffusion model; the average value and the bounds are reported.

| MODEL | ORIGINAL | | ROBUST | |
|---|---|---|---|---|
| ATTACK/EVALUATION | CLASS | TEXT-IMAGE | CLASS | TEXT-IMAGE |
| CLEAN | $1 \pm 0$ | $0.252 \pm 0.001$ | $1 \pm 0$ | $0.247 \pm 0.002$ |
| QF3 | $0.163 \pm 0.087$ | $0.143 \pm 0.003$ | $0.610 \pm 0.090$ | $0.194 \pm 0.008$ |
| RND-AP1 | $0.323 \pm 0.010$ | $0.175 \pm 0.002$ | $0.467 \pm 0.033$ | $0.192 \pm 0.001$ |
| QF5 | $0.190 \pm 0.060$ | $0.146 \pm 0.005$ | $0.513 \pm 0.087$ | $0.184 \pm 0.002$ |
| RND-AP2 | $0.317 \pm 0.083$ | $0.170 \pm 0.004$ | $0.470 \pm 0.030$ | $0.186 \pm 0.002$ |
| MMP1 | $0.597 \pm 0.037$ | $0.208 \pm 0.005$ | $0.697 \pm 0.047$ | $0.218 \pm 0.004$ |
| MMP2 | $0.263 \pm 0.063$ | $0.169 \pm 0.004$ | $0.427 \pm 0.027$ | $0.185 \pm 0.002$ |
| SDTAR1 | $0.750 \pm 0.050$ | $0.227 \pm 0.003$ | $0.777 \pm 0.027$ | $0.229 \pm 0.003$ |
| SDTAR2 | $0.667 \pm 0.067$ | $0.215 \pm 0.003$ | $0.683 \pm 0.007$ | $0.218 \pm 0.002$ |

**Experiments on a Different Version of the Stable Diffusion Model.** We conduct additional experiments in which we robustify an alternative version of the Stable Diffusion model on a set of labels from the CIFAR100 dataset. More precisely, we use the"runwayml/stable-diffusion-v1-4" (hug). The results for setting (S2) are presented in table 5. We notice that the robustified model offers improved performance over all attacks, similar to our previous experiments. It is therefore clear that the utility of the adversarial training method is not limited to a specific version of the underlying model.

## 5 Conclusion

In this study, we introduce an AT approach for SD. The AT method trains the system on a set of words, ensuring correct output even when the input undergoes certain adversarial modifications (e.g., "bicycle MJZM4"). However, our method has some limitations. First, it can robustify the model only against classes of attacks, specifically those attacks where the embeddings of the clean and adversarial prompts are close in a certain continuous text embedding space. Second, the applicability of the AT is limited to SD models. Therefore, in the future we plan to extend the applicability of the AT method to various text-to-image (T2I) systems beyond SD and to cover additional categories of adversarial attacks.

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

# A  Embeddings Of Clean And Adversarial Prompts On The Token and the Text Encoder Spaces

In Fig. 6, 7 we provide t-SNE plots of the embeddings of clean and adversarial prompts on the token embedding and text encoder embedding spaces. We observe that in the text encoder space the embeddings of the adversarial prompts cluster around the corresponding clean ones. On the other hand, there is no discernible pattern in the token embedding space. Overall, these figures reinforce the conclusions of the main text, which identified the text encoder embedding space as the suitable target on which the adversarial perturbations are generated.

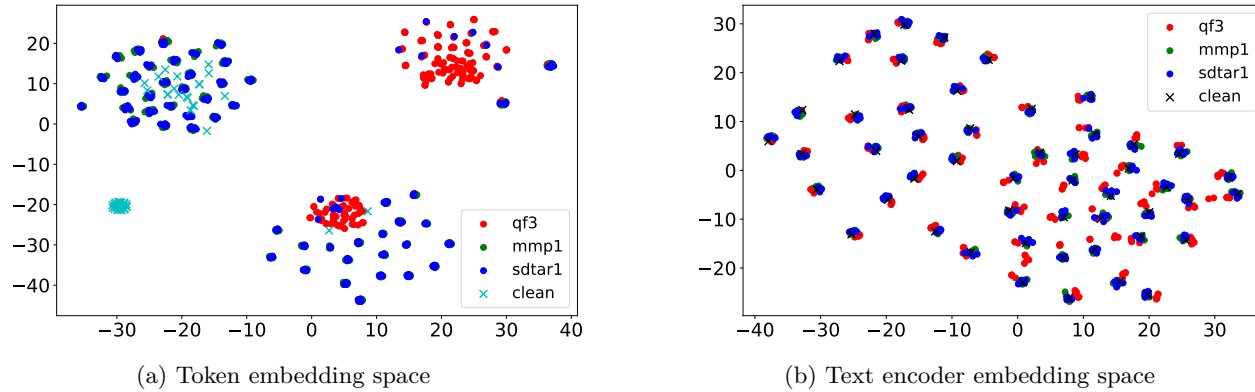

(a) Token embedding space                    (b) Text encoder embedding space

Figure 6: The embeddings of clean and adversarial prompts (QF3, MMP1, SDTAR1 attacks) in two different text embedding spaces available within SD.

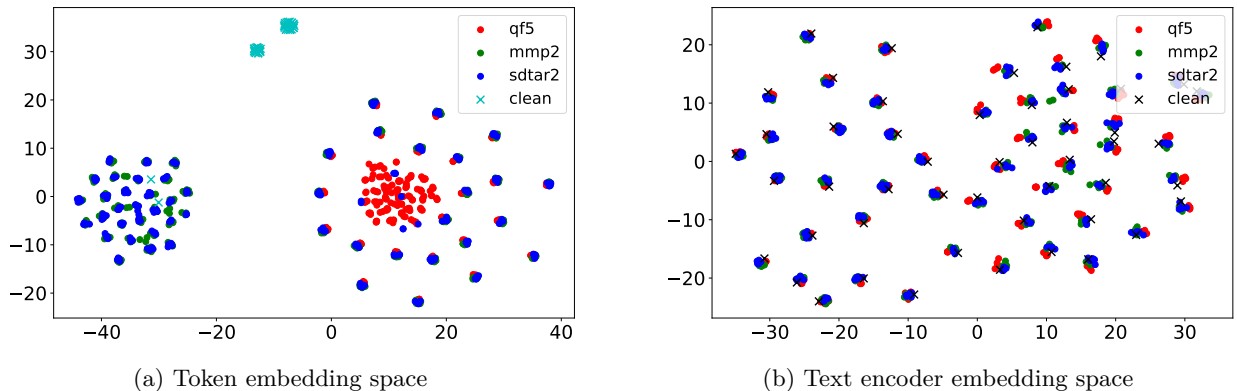

(a) Token embedding space                    (b) Text encoder embedding space

Figure 7: The embeddings of clean and adversarial prompts (QF5, MMP2, SDTAR2 attacks) in two different text embedding spaces available within SD.

# B  Evaluation of the UNet

In tables 6, 7 we present the results of running the MAT-SD algorithm with and without updating the UNet parameters (along with the parameters of the text encoder and the VAE decoder); the latter case corresponds to the proposed methodology we presented in the main text. We consider two different experiments depending on the base attacks we use to robustify the models (qf3/rnd-ap1 in table 6 and qf5/rnd-ap2 in table 7).

We note that while activating UNet parameters might result to improved performance in certain attacks, there is no consistent pattern. In fact, in several attacks it does not perform better than our current AT method and the attacks in which improvements are obtained are not known in advance. Given that training

Table 6: The performance of the robustified models, with and without updating the UNet parameters, against clean and adversarial prompts from the CIFAR100 dataset. For our evaluation we use the classification accuracy ("CLASS") and the text-image similarity ("TEXT-IMAGE") score of the outputs; higher scores indicate better performance. The adversarially trained model was robustified against the QF3 and RND-AP1 attacks and its performance is evaluated across the full set of attacks (setting S2). The results are averaged over 5 different random instances (seeds) of the Stable Diffusion model (v14); the average value and the full range of values attained over those different instances are reported.

| MODEL | ROBUST | | ROBUST (UNET) | |
|---|---|---|---|---|
| ATTACK/EVALUATION | CLASS | TEXT-IMAGE | CLASS | TEXT-IMAGE |
| CLEAN | $0.960 \pm 0.040$ | $0.237 \pm 0.002$ | $0.987 \pm 0.053$ | $0.245 \pm 0.001$ |
| QF3 | $0.853 \pm 0.063$ | $0.224 \pm 0.002$ | $0.647 \pm 0.047$ | $0.201 \pm 0.004$ |
| RND-AP1 | $0.333 \pm 0.083$ | $0.175 \pm 0.002$ | $0.357 \pm 0.060$ | $0.180 \pm 0.003$ |
| QF5 | $0.830 \pm 0.030$ | $0.221 \pm 0.001$ | $0.537 \pm 0.070$ | $0.189 \pm 0.002$ |
| RND-AP2 | $0.483 \pm 0.033$ | $0.185 \pm 0.004$ | $0.310 \pm 0.073$ | $0.174 \pm 0.007$ |
| MMP1 | $0.750 \pm 0.067$ | $0.220 \pm 0.003$ | $0.650 \pm 0.083$ | $0.209 \pm 0.002$ |
| MMP2 | $0.703 \pm 0.097$ | $0.207 \pm 0.002$ | $0.497 \pm 0.080$ | $0.191 \pm 0.002$ |
| SDTAR1 | $0.727 \pm 0.077$ | $0.216 \pm 0.003$ | $0.863 \pm 0.063$ | $0.231 \pm 0.004$ |
| SDTAR2 | $0.657 \pm 0.093$ | $0.209 \pm 0.004$ | $0.683 \pm 0.083$ | $0.216 \pm 0.005$ |

Table 7: The performance of the robustified models, with and without updating the UNet parameters, against clean and adversarial prompts from the CIFAR100 dataset. For our evaluation we use the classification accuracy ("CLASS") and the text-image similarity ("TEXT-IMAGE") score of the outputs; higher scores indicate better performance. The adversarially trained model was robustified against the QF5 and RND-AP1 attacks and its performance is evaluated across the full set of attacks (setting S2). The results are averaged over 5 different random instances (seeds) of the Stable Diffusion model (v14); the average value and the full range of values attained over those different instances are reported.

| MODEL | ROBUST | | ROBUST (UNET) | |
|---|---|---|---|---|
| ATTACK/EVALUATION | CLASS | TEXT-IMAGE | CLASS | TEXT-IMAGE |
| CLEAN | $0.960 \pm 0.093$ | $0.257 \pm 0.001$ | $0.987 \pm 0.053$ | $0.244 \pm 0.004$ |
| QF5 | $0.613 \pm 0.087$ | $0.218 \pm 0.004$ | $0.600 \pm 0.100$ | $0.199 \pm 0.010$ |
| RND-AP2 | $0.367 \pm 0.067$ | $0.195 \pm 0.009$ | $0.327 \pm 0.060$ | $0.178 \pm 0.004$ |
| QF3 | $0.670 \pm 0.063$ | $0.226 \pm 0.007$ | $0.613 \pm 0.047$ | $0.202 \pm 0.008$ |
| RND-AP1 | $0.300 \pm 0.033$ | $0.194 \pm 0.004$ | $0.343 \pm 0.107$ | $0.187 \pm 0.003$ |
| MMP1 | $0.707 \pm 0.093$ | $0.236 \pm 0.002$ | $0.660 \pm 0.073$ | $0.217 \pm 0.004$ |
| MMP2 | $0.567 \pm 0.033$ | $0.218 \pm 0.003$ | $0.567 \pm 0.100$ | $0.201 \pm 0.004$ |
| SDTAR1 | $0.810 \pm 0.077$ | $0.244 \pm 0.001$ | $0.747 \pm 0.064$ | $0.229 \pm 0.005$ |
| SDTAR2 | $0.770 \pm 0.037$ | $0.239 \pm 0.001$ | $0.767 \pm 0.050$ | $0.226 \pm 0.004$ |

the UNet incurs significant costs, making the AT process harder to tune and slower, without providing clear benefits, we have decided against incorporating it into our method.

## C  Evaluation of the Output Sensitivity Loss Term

In tables 8, 9 we present the results of running the MAT-SD algorithm with and without incorporating the output sensitivity loss term (along with the text-image similarity and classification loss terms); the latter case corresponds to the proposed methodology we presented in the main text. We consider two different experiments depending on the base attacks we use to robustify the models (qf3/rnd-ap1 in table 8 and qf5/rnd-ap2 in table 9).

Table 8: The performance of the robustified models, with and without the incorporation of the output sensitivity loss term, against clean and adversarial prompts from the CIFAR100 dataset. For our evaluation we use the classification accuracy ("CLASS") and the text-image similarity ("TEXT-IMAGE") score of the outputs; higher scores indicate better performance. The adversarially trained model was robustified against the QF3 and RND-AP1 attacks and its performance is evaluated across the full set of attacks (setting S2). The results are averaged over 5 different random instances (seeds) of the Stable Diffusion model (v14); the average value and the full range of values attained over those different instances are reported.

| MODEL | ROBUST | | ROBUST (OUT SENSITIVITY) | |
|---|---|---|---|---|
| ATTACK/EVALUATION | CLASS | TEXT-IMAGE | CLASS | TEXT-IMAGE |
| CLEAN | $0.960 \pm 0.040$ | $0.237 \pm 0.002$ | $1.0 \pm 0.0$ | $0.244 \pm 0.001$ |
| QF3 | $0.853 \pm 0.063$ | $0.224 \pm 0.002$ | $0.673 \pm 0.043$ | $0.218 \pm 0.005$ |
| RND-AP1 | $0.333 \pm 0.083$ | $0.175 \pm 0.002$ | $0.250 \pm 0.033$ | $0.167 \pm 0.003$ |
| QF5 | $0.830 \pm 0.030$ | $0.221 \pm 0.001$ | $0.667 \pm 0.100$ | $0.211 \pm 0.007$ |
| RND-AP2 | $0.483 \pm 0.033$ | $0.185 \pm 0.004$ | $0.243 \pm 0.040$ | $0.165 \pm 0.005$ |
| MMP1 | $0.750 \pm 0.067$ | $0.220 \pm 0.003$ | $0.657 \pm 0.043$ | $0.212 \pm 0.004$ |
| MMP2 | $0.703 \pm 0.097$ | $0.207 \pm 0.002$ | $0.457 \pm 0.093$ | $0.191 \pm 0.003$ |
| SDTAR1 | $0.727 \pm 0.077$ | $0.216 \pm 0.003$ | $0.773 \pm 0.140$ | $0.227 \pm 0.008$ |
| SDTAR2 | $0.657 \pm 0.093$ | $0.209 \pm 0.004$ | $0.700 \pm 0.150$ | $0.221 \pm 0.005$ |

Table 9: The performance of the robustified models, with and without the incorporation of the output sensitivity loss term, against clean and adversarial prompts from the CIFAR100 dataset. For our evaluation we use the classification accuracy ("CLASS") and the text-image similarity ("TEXT-IMAGE") score of the outputs; higher scores indicate better performance. The adversarially trained model was robustified against the QF5 and RND-AP2 attacks and its performance is evaluated across the full set of attacks (setting S2). The results are averaged over 5 different random instances (seeds) of the Stable Diffusion model (v14); the average value and the full range of values attained over those different instances are reported.

| MODEL | ROBUST | | ROBUST (OUT SENSITIVITY) | |
|---|---|---|---|---|
| ATTACK/EVALUATION | CLASS | TEXT-IMAGE | CLASS | TEXT-IMAGE |
| CLEAN | $0.960 \pm 0.093$ | $0.257 \pm 0.001$ | $1.0 \pm 0.0$ | $0.247 \pm 0.0025$ |
| QF5 | $0.613 \pm 0.087$ | $0.218 \pm 0.004$ | $0.733 \pm 0.050$ | $0.210 \pm 0.002$ |
| RND-AP2 | $0.367 \pm 0.067$ | $0.195 \pm 0.009$ | $0.457 \pm 0.040$ | $0.185 \pm 0.003$ |
| QF3 | $0.670 \pm 0.063$ | $0.226 \pm 0.007$ | $0.730 \pm 0.030$ | $0.213 \pm 0.003$ |
| RND-AP1 | $0.300 \pm 0.033$ | $0.194 \pm 0.004$ | $0.357 \pm 0.027$ | $0.177 \pm 0.002$ |
| MMP1 | $0.707 \pm 0.093$ | $0.236 \pm 0.002$ | $0.667 \pm 0.050$ | $0.219 \pm 0.002$ |
| MMP2 | $0.567 \pm 0.033$ | $0.218 \pm 0.003$ | $0.480 \pm 0.063$ | $0.189 \pm 0.004$ |
| SDTAR1 | $0.810 \pm 0.077$ | $0.244 \pm 0.001$ | $0.810 \pm 0.090$ | $0.230 \pm 0.004$ |
| SDTAR2 | $0.770 \pm 0.037$ | $0.239 \pm 0.001$ | $0.723 \pm 0.127$ | $0.222 \pm 0.003$ |

We note that the incorporation of the output sensitivity loss leads to improvements in certain attacks compared to the main method. However, similar to the UNet case, these improvements do not occur across all attacks, and there is no clear pattern regarding when these improvements appear (i.e., under which base attack set) or on which specific attacks. As the introduction of a third loss term incurs additional costs (e.g., hyperparameter tuning becomes more challenging), we decided not to include this loss in our main method.

## D  The Proposed MAT-SD Algorithm

We restate the convergence theorem from the main text and provide the proof.

**Theorem D.1.** *Suppose that the functions $f, h, g, e, c, \ell_{sim}, \ell_{cls}$ are twice continuously differentiable, have Lipschitz continuous and bounded gradients, and Lipschitz continuous and bounded Hessians/Jacobians; also suppose that $\mathcal{L}$ has a lower bound. We linearize the loss $\mathcal{L}$ with respect to $\delta$ (around $\delta = 0$), i.e.,*

$$\widetilde{\mathcal{L}}((\theta, \phi), \delta) = \mathcal{L}((\theta, \phi), 0) + \nabla_\delta \mathcal{L}((\theta, \phi), 0)\, \delta,$$

*and apply the HiBSA algorithm (Lu et al., 2020) to solve the min-max problem 1:*

$$\min_{\theta, \phi} \max_{\delta \in \Delta} \widetilde{\mathcal{L}}((\theta, \phi), \delta). \tag{7}$$

*Then the iterates of HiBSA converge to a stationary solution of the above min-max problem.*

*Proof.* To prove the convergence of the HiBSA algorithm (Lu et al., 2020) to a stationary solution of problem 7, we need to ensure that HiBSA's assumptions hold for the objective $\widetilde{\mathcal{L}}((\theta, \phi), \delta)$.

First, we note that the objective $\widetilde{\mathcal{L}}((\theta, \phi), \delta)$ is differentiable as a composition of twice continuously differentiable functions. Second, the linearized objective $\widetilde{\mathcal{L}}((\theta, \phi), \delta) = \mathcal{L}((\theta, \phi), 0) + \nabla_\delta \mathcal{L}((\theta, \phi), 0)\, \delta$ is non-convex in $(\theta, \delta)$ and concave (linear) in $\delta$. Then, it suffices to show that the objective $\widetilde{\mathcal{L}}((\theta, \phi), \delta)$ has Lipschitz continuous gradients. We have that

$$\nabla_\theta \widetilde{\mathcal{L}}((\theta, \phi), \delta) = \nabla_\theta \mathcal{L}((\theta, \phi), 0) + \nabla^2_{\theta\delta} \mathcal{L}((\theta, \phi), 0)\, \delta$$
$$\nabla_\phi \widetilde{\mathcal{L}}((\theta, \phi), \delta) = \nabla_\phi \mathcal{L}((\theta, \phi), 0) + \nabla^2_{\phi\delta} \mathcal{L}((\theta, \phi), 0)\, \delta$$
$$\nabla_\delta \widetilde{\mathcal{L}}((\theta, \phi), \delta) = \nabla_\delta \mathcal{L}((\theta, \phi), 0)$$

Therefore, we need to show the Lipschitz continuity of every term in the rhs of the above expressions. We split this work into two parts. First, we establish the desired property for the first-order terms $\nabla_\theta \mathcal{L}((\theta, \phi), 0)$, $\nabla_\phi \mathcal{L}((\theta, \phi), 0)$, $\nabla_\delta \mathcal{L}((\theta, \phi), 0)$. Then, we establish the same property for the second-order terms $\nabla^2_{\theta\delta} \mathcal{L}((\theta, \phi), 0)$, $\nabla^2_{\phi\delta} \mathcal{L}((\theta, \phi), 0)$. Finally, in part 3 we combine the results from the previous two parts to complete the proof.

**Part 1: Lipschitz continuity of $\nabla_\theta \mathcal{L}((\theta, \phi), 0)$, $\nabla_\phi \mathcal{L}((\theta, \phi), 0)$, $\nabla_\delta \mathcal{L}((\theta, \phi), 0)$**

First, let us focus on the left terms of $\nabla_\theta \mathcal{L}((\theta, \phi), 0)$, $\nabla_\phi \mathcal{L}((\theta, \phi), 0)$, $\nabla_\delta \mathcal{L}((\theta, \phi), 0)$. We want to show that each of those terms is Lipschitz continuous. To do so we compute the gradients of all the involved expressions. For simplicity assume that the Lipschitz boundedness and continuous gradient constants are $\overline{L}$ and $L$, respectively, i.e., they are the same across all the involved functions. Then, for the gradients of the involved expression the following hold.

1. $\mathcal{L}\left((\theta, \phi), \delta; \{x_i\}_{i=1}^N\right) = -\sum_{i=1}^N \mathcal{L}_i\left((\theta, \phi), \delta; x_i\right)$

$$\nabla \mathcal{L}\left((\theta, \phi), \delta; \{x_i\}_{i=1}^N\right) = -\sum_{i=1}^N \nabla \mathcal{L}_i\left((\theta, \phi), \delta; x_i\right)$$

2. $\mathcal{L}_i\left((\theta, \phi), \delta; x_i\right) = \lambda_1 \cdot \ell_{\text{sim}}\left(\widehat{f}(x_i), w_i(\theta, \phi, \delta)\right) + \lambda_2 \cdot \ell_{\text{cls}}\left(\bar{x}_i, u_i(\theta, \phi, \delta)\right)$

$$\nabla_{(\theta, \phi)} \mathcal{L}_i\left((\theta, \phi), \delta; x_i\right) = \nabla_{(\theta, \phi)} w_i(\theta, \phi, \delta) \nabla_2 \ell_{\text{sim}}\left(\widehat{f}(x_i), w_i(\theta, \phi, \delta)\right) + \nabla_{(\theta, \phi)} u_i(\theta, \phi, \delta) \nabla_2 \ell_{\text{cls}}\left(\bar{x}_i, u_i(\theta, \phi, \delta)\right)$$

$$\nabla_\delta \mathcal{L}_i\left((\theta, \phi), \delta; x_i\right) = \nabla_\delta w_i(\theta, \phi, \delta) \nabla_2 \ell_{\text{sim}}\left(\widehat{f}(x_i), w_i(\theta, \phi, \delta)\right) + \nabla_\delta u_i(\theta, \phi, \delta) \nabla_2 \ell_{\text{cls}}\left(\bar{x}_i, u_i(\theta, \phi, \delta)\right)$$

3. $w_i(\theta, \phi, \delta) := e(z_i(\theta, \phi, \delta))$

$$\nabla_\theta w_i(\theta, \phi, \delta) = \nabla_\theta z_i(\theta, \phi, \delta) \nabla e(z_i(\theta, \phi, \delta)), \nabla_\phi w_i(\theta, \phi, \delta) = \nabla_\phi z_i(\theta, \phi, \delta) \nabla e(z_i(\theta, \phi, \delta))$$
$$\nabla_\delta w_i(\theta, \phi, \delta) = \nabla_\delta z_i(\theta, \phi, \delta) \nabla e(z_i(\theta, \phi, \delta))$$

4. $u_i(\theta, \phi, \delta) := c(z_i(\theta, \phi, \delta))$

$$\nabla_\theta u_i(\theta, \phi, \delta) = \nabla_\theta z_i(\theta, \phi, \delta) \nabla c(z_i(\theta, \phi, \delta)), \nabla_\phi u_i(\theta, \phi, \delta) = \nabla_\phi z_i(\theta, \phi, \delta) \nabla c(z_i(\theta, \phi, \delta))$$
$$\nabla_\delta u_i(\theta, \phi, \delta) = \nabla_\delta z_i(\theta, \phi, \delta) \nabla c(z_i(\theta, \phi, \delta))$$

5. $z_i(\theta, \phi, \delta) := g(p(\theta, \delta); \phi); p(\theta, \delta) := h(y_i(\theta, \delta))$

$$\nabla_\theta z_i(\theta, \phi, \delta) = \nabla_\theta p(\theta, \delta) \nabla_1 g(p(\theta, \delta); \phi), \nabla_\phi z_i(\theta, \phi, \delta) = \nabla_2 g(p(\theta, \delta); \phi)$$
$$\nabla_\delta z_i(\theta, \phi, \delta) = \nabla_\delta p(\theta, \delta) \nabla_1 g(p(\theta, \delta); \phi)$$

6. $p(\theta, \delta) := h(y_i(\theta, \delta))$

$$\nabla_\theta p(\theta, \delta) = \nabla_\theta y_i(\theta, \delta) \nabla h(y_i(\theta, \delta)), \nabla_\phi h y(\theta, \delta) = 0$$
$$\nabla_\delta p(\theta, \delta) = \nabla_\delta y_i(\theta, \delta) \nabla h(y_i(\theta, \delta))$$

7. $y_i(\theta, \delta) := f(x_i; \theta) + \delta$

$$\nabla_\theta y_i(\theta, \delta) = \nabla_\theta f(x_i; \theta), \nabla_\phi y_i(\theta, \delta) = 0$$
$$\nabla_\delta y_i(\theta, \delta) = I$$

Starting with the expression in item 7 we can show that under the imposed assumption $y_i(\theta, \delta)$ has Lipschitz and bounded gradients. Specifically, the gradient boundedness of $f$ implies the boundedness of $\|\nabla_\theta y_i(\theta, \delta)\| \le \overline{L}$; the boundedness of $\|\nabla_\phi y_i(\theta, \delta)\|$ and $\|\nabla_\delta y_i(\theta, \delta)\|$ follows trivially.

Moreover, for the Lipschitz gradient property we have

$$\|\nabla_\theta y_i(\theta_1, \delta_1) - \nabla_\theta y_i(\theta_2, \delta_2)\| = \|\nabla_\theta f(x_i; \theta_1) - \nabla_\theta f(x_i; \theta_2)\| \le L\|\theta_1 - \theta_2\| \le L\|(\theta_1, \phi_1, \delta_1) - (\theta_2, \phi_2, \delta_2)\|$$
$$\|\nabla_\phi y_i(\theta_1, \delta_1) - \nabla_\phi y_i(\theta_2, \delta_2)\| = 0 \le L\|(\theta_1, \phi_1, \delta_1) - (\theta_2, \phi_2, \delta_2)\|$$
$$\|\nabla_\delta y_i(\theta_1, \delta_1) - \nabla_\delta y_i(\theta_2, \delta_2)\| = 0 \le L\|(\theta_1, \phi_1, \delta_1) - (\theta_2, \phi_2, \delta_2)\|, \tag{8}$$

where the first expression follows from the Lipschitz gradient property of $f$.

Next, we consider the expression in item 6.

$$\|\nabla_\theta p(\theta, \delta)\| = \|\nabla_\theta y_i(\theta, \delta) \nabla h(y_i(\theta, \delta))\| \le \|\nabla_\theta y_i(\theta, \delta)\| \|\nabla h(y_i(\theta, \delta))\| \le \overline{L}^2$$
$$\|\nabla_\phi p(\theta, \delta)\| = 0$$
$$\|\nabla_\delta p(\theta, \delta)\| = \|\nabla_\delta y_i(\theta, \delta) \nabla h(y_i(\theta, \delta))\| \le \|\nabla_\delta y_i(\theta, \delta)\| \|\nabla h(y_i(\theta, \delta))\| \le \overline{L}^2$$

Therefore, the gradients are bounded. Moreover, using the bounded and Lipschitz gradient of $h$ (by assumption) and $y_i$ (from the derivations in 8), we obtain the following:

$\|\nabla_\theta p(\theta_1, \delta_1) - \nabla_\theta p(\theta_2, \delta_2)\|$
$= \|\nabla_\theta y_i(\theta_1, \delta_1) \nabla h(y_i(\theta_1, \delta_1)) - \nabla_\theta y_i(\theta_2, \delta_2) \nabla h(y_i(\theta_2, \delta_2))\| =$
$= \|\nabla_\theta y_i(\theta_1, \delta_1) \nabla h(y_i(\theta_1, \delta_1)) - \nabla_\theta y_i(\theta_2, \delta_2) \nabla h(y_i(\theta_1, \delta_1)) + \nabla_\theta y_i(\theta_2, \delta_2) \nabla h(y_i(\theta_1, \delta_1)) - \nabla_\theta y_i(\theta_2, \delta_2) \nabla h(y_i(\theta_2, \delta_2))\|$
$\le \|\nabla_\theta y_i(\theta_1, \delta_1) - \nabla_\theta y_i(\theta_2, \delta_2)\| \|\nabla h(y_i(\theta_1, \delta_1))\| + \|\nabla_\theta y_i(\theta_2, \delta_2)\| \|\nabla h(y_i(\theta_1, \delta_1)) - \nabla h(y_i(\theta_2, \delta_2))\|$
$\le \overline{L} L \|(\theta_1, \phi_1, \delta_1) - (\theta_2, \phi_2, \delta_2)\|$
$\|\nabla_\phi p(\theta_1, \delta_1) - \nabla_\theta p(\theta_2, \delta_2)\| = 0 \le L\|(\theta_1, \phi_1, \delta_1) - (\theta_2, \phi_2, \delta_2)\|$
$\|\nabla_\delta p(\theta_1, \delta_1) - \nabla_\delta p(\theta_2, \delta_2)\|$
$= \|\nabla_\delta y_i(\theta_1, \delta_1) \nabla h(y_i(\theta_1, \delta_1)) - \nabla_\delta y_i(\theta_2, \delta_2) \nabla h(y_i(\theta_2, \delta_2))\| =$
$= \|\nabla_\delta y_i(\theta_1, \delta_1) \nabla h(y_i(\theta_1, \delta_1)) - \nabla_\delta y_i(\theta_2, \delta_2) \nabla h(y_i(\theta_1, \delta_1)) + \nabla_\delta y_i(\theta_2, \delta_2) \nabla h(y_i(\theta_1, \delta_1)) - \nabla_\delta y_i(\theta_2, \delta_2) \nabla h(y_i(\theta_2, \delta_2))\|$

$$\leq \|\nabla_\delta y_i(\theta_1, \delta_1) - \nabla_\delta y_i(\theta_2, \delta_2)\|\|\nabla h(y_i(\theta_1, \delta_1))\| + \|\nabla_\delta y_i(\theta_2, \delta_2)\|\|\nabla h(y_i(\theta_1, \delta_1)) - \nabla h(y_i(\theta_2, \delta_2))\|$$

$$\leq \overline{L}L\|(\theta_1, \phi_1, \delta_1) - (\theta_2, \phi_2, \delta_2)\|$$

The above prove the Lipschitz gradient property of $p$.

Following the same reasoning we can show the bounded and Lipschitz gradient property of the expressions in items 1-5, i.e., we establish the aforementioned properties for $z_i(\theta, \phi, \delta)$, $w_i(\theta, \phi, \delta)$, $u_i(\theta, \phi, \delta)$, $\mathcal{L}_i((\theta, \phi), \delta; x_i)$ and $\mathcal{L}((\theta, \phi), 0)$.

**Part 2: Lipschitz continuity and boundedness of $\nabla_{\theta\delta}^2 \mathcal{L}((\theta, \phi), 0)$, $\nabla_{\phi\delta}^2 \mathcal{L}((\theta, \phi), 0)$**

Next, we move into establishing the bounded and Lipschitz gradient properties for the terms $\nabla_{\theta\delta}^2 \mathcal{L}((\theta, \phi), 0)$ and $\nabla_{\phi\delta}^2 \mathcal{L}((\theta, \phi), 0)$; we actually focus only on the former term as the derivations are very similar in both of them. Also, to simplify the analysis below we consider the notation $\pi = (\theta, \phi, \delta)$. Similar to part 1 we split the derivation into seven items in order to facilitate the presentation.

1. $\mathcal{L}\left(\pi; \{x_i\}_{i=1}^N\right) = -\sum_{i=1}^N \mathcal{L}_i(\pi; x_i)$

$$\nabla_{\theta\delta}^2 \mathcal{L}\left(\pi; \{x_i\}_{i=1}^N\right) = -\sum_{i=1}^N \nabla_{\theta\delta}^2 \mathcal{L}_i(\pi; x_i)$$

2. $\mathcal{L}_i(\pi; x_i) = \lambda_1 \cdot \ell_{\text{sim}}\left(\widehat{f}(x_i), w_i(\pi)\right) + \lambda_2 \cdot \ell_{\text{cls}}(\bar{x}_i, u_i(\pi))$

   We notice that we can rewrite the gradient of this expression in the following way

   $$\nabla_\delta \mathcal{L}_i(\pi; x_i)$$
   $$= \lambda_1 \nabla_\delta w_i(\pi) \nabla_2 \ell_{\text{sim}}\left(\widehat{f}(x_i), w_i(\pi)\right) + \lambda_2 \nabla_\delta u_i(\pi) \nabla_2 \ell_{\text{cls}}(\bar{x}_i, u_i(\pi))$$
   $$= \lambda_1 \sum_j [\nabla_\delta w_i(\pi)]_{:,j} \left[\nabla_2 \ell_{\text{sim}}\left(\widehat{f}(x_i), w_i(\pi)\right)\right]_j + \lambda_2 \sum_j [\nabla_\delta u_i(\pi)]_{:,j} [\nabla_2 \ell_{\text{cls}}(\bar{x}_i, u_i(\pi))]_j,$$

   where we denote with the notation $[\cdot]_{:,i}$ the $i$th column of the respective matrix and with the notation $[\cdot]_i$ the $i$th element of the respective vector.

   Then, we can differentiate the above expression with respect to $\theta$.

   $$\nabla_{\theta\delta}^2 \mathcal{L}_i(\pi; x_i) = \lambda_1 \sum_j \left\{ \nabla_\theta [\nabla_\delta w_i(\pi)]_{:,j} \left[\nabla_2 \ell_{\text{sim}}\left(\widehat{f}(x_i), w_i(\pi)\right)\right]_j + \nabla_\theta w_i(\pi) \left[\nabla_{22} \ell_{\text{sim}}\left(\widehat{f}(x_i), w_i(\pi)\right)\right]_{:,j} [\nabla_\delta w_i(\pi)]_{:,j}^T \right\}$$
   $$+ \lambda_2 \sum_j \left\{ \nabla_\theta [\nabla_\delta u_i(\pi)]_{:,j} [\nabla_2 \ell_{\text{cls}}(\bar{x}_i, u_i(\pi))]_j + \nabla_\theta u_i(\theta, \phi, \delta) [\nabla_{22} \ell_{\text{cls}}(\bar{x}_i, u_i(\pi))]_{:,j} [\nabla_\delta u_i(\pi)]_{:,j}^T \right\}$$

   Let's assume (temporarily) that $w_i(\pi)$ and $u_i(\pi)$ have Lipschitz continuous and bounded Jacobians. Then, for the boundedness of the Jacobian we have the following.

   $$\|\nabla_{\theta\delta}^2 \mathcal{L}_i(\pi; x_i)\|$$
   $$\leq \lambda_1 \sum_j \left\{ \left\|\nabla_\theta [\nabla_\delta w_i(\pi)]_{:,j}\right\| \left\|\left[\nabla_2 \ell_{\text{sim}}\left(\widehat{f}(x_i), w_i(\pi)\right)\right]_j\right\| \right.$$
   $$+ \|\nabla_\theta w_i(\pi)\| \left\|\left[\nabla_{22} \ell_{\text{sim}}\left(\widehat{f}(x_i), w_i(\pi)\right)\right]_{:,j}\right\| \left\|[\nabla_\delta w_i(\pi)]_{:,j}^T\right\| \right\}$$
   $$+ \lambda_2 \sum_j \left\{ \left\|\nabla_\theta [\nabla_\delta u_i(\pi)]_{:,j}\right\| \left\|[\nabla_2 \ell_{\text{cls}}(\bar{x}_i, u_i(\pi))]_j\right\| + \|\nabla_\theta u_i(\pi)\| \left\|[\nabla_{22} \ell_{\text{cls}}(\bar{x}_i, u_i(\pi))]_{:,j}\right\| \left\|[\nabla_\delta u_i(\pi)]_{:,j}^T\right\| \right\}$$
   
   $$\tag{9}$$

The above expression is bounded as all of the involved expressions are bounded. Specifically, the boundedness of $\nabla_\theta \left[\nabla_\delta w_i(\pi)\right]_{:,j}$ and $\nabla_\theta \left[\nabla_\delta u_i(\theta,\phi,\delta)\right]_{:,j}$ follow from the (temporary) assumptions we imposed right above (to be proven below). The boundedness of $[\nabla_\delta w_i(\pi)]_{:,j}$, $[\nabla_\delta u_i(\pi)]_{:,j}$, $\nabla_\theta w_i(\pi)$, $\nabla_\theta u_i(\pi)$ follow from the derivations of Part 1. The boundedness of the remaining terms, i.e, $\left[\nabla_2 \ell_{\text{sim}}\left(\widehat{f}(x_i), w_i(\pi)\right)\right]_j$, $\left[\nabla_{22}\ell_{\text{sim}}\left(\widehat{f}(x_i), w_i(\pi)\right)\right]_{:,j}$, $[\nabla_2 \ell_{\text{cls}}(\bar{x}_i, u_i(\pi))]_j$ and $[\nabla_{22}\ell_{\text{cls}}(\bar{x}_i, u_i(\pi))]_{:,j}$ follows from the theorem's assumptions.

Moreover, for the Lipschitz gradient continuity we consider the following derivation

$$\|\nabla^2_{\theta\delta}\mathcal{L}_i(\pi_1; x_i) - \nabla^2_{\theta\delta}\mathcal{L}_i(\pi_2; x_i)\|$$

$$\leq \lambda_1 \sum_j \left\{ \left\|\nabla_\theta\left[\nabla_\delta w_i(\pi_1)\right]_{:,j}\right\| \left[\left\|\left[\nabla_2\ell_{\text{sim}}\left(\widehat{f}(x_i), w_i(\pi_1)\right)\right]_j - \left[\nabla_2\ell_{\text{sim}}\left(\widehat{f}(x_i), w_i(\pi_2)\right)\right]_j\right\|\right] \right.$$

$$+ \left\|\nabla_\theta\left[\nabla_\delta w_i(\pi_1)\right]_{:,j} - \nabla_\theta\left[\nabla_\delta w_i(\pi_2)\right]_{:,j}\right\| \left\|\left[\nabla_2\ell_{\text{sim}}\left(\widehat{f}(x_i), w_i(\pi_2)\right)\right]_j\right\|$$

$$+ \left\|\nabla_\theta w_i(\pi_1)\left[\nabla_{22}\ell_{\text{sim}}\left(\widehat{f}(x_i), w_i(\pi_1)\right)\right]_{:,j}\right\| \left\|[\nabla_\delta w_i(\pi_1)]^T_{:,j} - [\nabla_\delta w_i(\pi_2)]^T_{:,j}\right\|$$

$$+ \left\|\nabla_\theta w_i(\pi_1)[\nabla_\delta w_i(\pi_2)]^T_{:,j}\right\| \left\|\left[\nabla_{22}\ell_{\text{sim}}\left(\widehat{f}(x_i), w_i(\pi_1)\right)\right]_{:,j} - \left[\nabla_{22}\ell_{\text{sim}}\left(\widehat{f}(x_i), w_i(\pi_2)\right)\right]_{:,j}\right\|$$

$$\left. + \left\|\left[\nabla_{22}\ell_{\text{sim}}\left(\widehat{f}(x_i), w_i(\pi_2)\right)\right]_{:,j}[\nabla_\delta w_i(\pi_2)]^T_{:,j}\right\| \|\nabla_\theta w_i(\pi_1) - \nabla_\theta w_i(\pi_2)\| \right\}$$

$$\leq \lambda_2 \sum_j \left\{ \left\|\nabla_\theta\left[\nabla_\delta u_i(\pi_1)\right]_{:,j}\right\| \left[\left\|\left[\nabla_2\ell_{\text{sim}}\left(\widehat{f}(x_i), u_i(\pi_1)\right)\right]_j - \left[\nabla_2\ell_{\text{sim}}\left(\widehat{f}(x_i), u_i(\pi_2)\right)\right]_j\right\|\right] \right.$$

$$+ \left\|\nabla_\theta\left[\nabla_\delta u_i(\pi_1)\right]_{:,j} - \nabla_\theta\left[\nabla_\delta u_i(\pi_2)\right]_{:,j}\right\| \left\|\left[\nabla_2\ell_{\text{sim}}\left(\widehat{f}(x_i), u_i(\pi_2)\right)\right]_j\right\|$$

$$+ \left\|\nabla_\theta u_i(\pi_1)\left[\nabla_{22}\ell_{\text{sim}}\left(\widehat{f}(x_i), u_i(\pi_1)\right)\right]_{:,j}\right\| \left\|[\nabla_\delta u_i(\pi_1)]^T_{:,j} - [\nabla_\delta u_i(\pi_2)]^T_{:,j}\right\|$$

$$+ \left\|\nabla_\theta u_i(\pi_1)[\nabla_\delta u_i(\pi_2)]^T_{:,j}\right\| \left\|\left[\nabla_{22}\ell_{\text{sim}}\left(\widehat{f}(x_i), u_i(\pi_1)\right)\right]_{:,j} - \left[\nabla_{22}\ell_{\text{sim}}\left(\widehat{f}(x_i), u_i(\pi_2)\right)\right]_{:,j}\right\|$$

$$\left. + \left\|\left[\nabla_{22}\ell_{\text{sim}}\left(\widehat{f}(x_i), u_i(\pi_2)\right)\right]_{:,j}[\nabla_\delta u_i(\pi_2)]^T_{:,j}\right\| \|\nabla_\theta u_i(\pi_1) - \nabla_\theta u_i(\pi_2)\| \right\}$$

$$\leq \hat{L}_1 \|\pi_1 - \pi_2\|,$$

for some proper Lipschitz constant $\hat{L}_1 > 0$. In the above, all the expressions (within the norms) are either bounded or have Lipschitz gradients (this follows either from the assumptions or from the derivations of Part 1).

Then, it suffices to show that $w_i(\pi)$ and $u_i(\pi)$ have Lipschitz continuous and bounded Jacobians (the items 3 and 4 below, respectively).

3. $w_i(\pi) := e(z_i(\pi))$

   Consider, for instance, the Jacobian of $w_i(\pi)$. It can be rewritten as follows:

$$\nabla_\theta\left[\nabla_\delta w_i(\pi)\right]_{:,j} = \nabla_\theta\left[\nabla_\delta z_i(\pi)\nabla c(z_i(\pi))\right]_{:,j}$$

$$= \nabla_\theta\left\{\nabla_\delta z_i(\pi)\left[\nabla c(z_i(\pi))\right]_{:,j}\right\}$$

$$= \nabla_\theta\left\{\sum_k \left[\nabla_\delta z_i(\pi)\right]_{:,k}\left[\nabla c(z_i(\pi))\right]_{k,j}\right\}$$

$$= \sum_k \left\{ \nabla_\theta \left[ \nabla_\delta z_i(\pi) \right]_{:,k} \left[ \nabla c(z_i(\pi)) \right]_{k,j} + \nabla_\theta z_i(\pi) \nabla \left[ \nabla c(z_i(\pi)) \right]_{k,j} \left[ \nabla_\delta z_i(\pi) \right]_{:,k}^T \right\}$$

The structure of the above Jacobian is the same as the structure of the terms in eq. 9. Therefore, we can prove the Lipschitz boundedness and continuity of $\nabla_\theta \left[ \nabla_\delta w_i(\pi) \right]_{:,j}$ by following the same reasoning used for $\nabla^2_{\theta\delta} \mathcal{L}_i(\pi; x_i)$. Similar to this case our derivation will depend on the assumption that $\nabla_\theta \left[ \nabla_\delta z_i(\pi) \right]_{:,j}$ (the item 5 below) has Lipschitz continuous and bounded Jacobians.

4. $u_i(\pi) := c(z_i(\pi))$

   Similarly to the case of the above item, we have the following expression for the Jacobian of $u_i(\pi)$

$$\nabla_\theta \left[ \nabla_\delta u_i(\pi) \right]_{:,j} = \nabla_\theta \left[ \nabla_\delta z_i(\pi) \nabla c(z_i(\pi)) \right]_{:,j}$$
$$= \nabla_\theta \left\{ \nabla_\delta z_i(\pi) \left[ \nabla c(z_i(\pi)) \right]_{:,j} \right\}$$
$$= \nabla_\theta \left\{ \sum_k \left[ \nabla_\delta z_i(\pi) \right]_{:,k} \left[ \nabla c(z_i(\pi)) \right]_{k,j} \right\}$$
$$= \sum_k \left\{ \nabla_\theta \left[ \nabla_\delta z_i(\pi) \right]_{:,k} \left[ \nabla c(z_i(\pi)) \right]_{k,j} + \nabla_\theta z_i(\pi) \nabla \left[ \nabla c(z_i(\pi)) \right]_{k,j} \left[ \nabla_\delta z_i(\pi) \right]_{:,k}^T \right\}$$

5. $z_i(\pi) := g(p(\pi); \phi); p(\pi) := h(y_i(\pi))$

   Similarly to the case of the above item, we have the following expression for the Jacobian of $z_i(\pi)$

$$\nabla_\theta \left[ \nabla_\delta z_i(\pi) \right]_{:,k} = \nabla_\theta \left[ \nabla_\delta p(\pi) \nabla g(p(\pi)) \right]_{:,k}$$
$$= \nabla_\theta \left\{ \nabla_\delta p(\pi) \left[ \nabla g(p(\pi)) \right]_{:,k} \right\}$$
$$= \nabla_\theta \left\{ \sum_l \left[ \nabla_\delta p(\pi) \right]_{:,l} \left[ \nabla g(p(\pi)) \right]_{l,k} \right\}$$
$$= \sum_l \left\{ \nabla_\theta \left[ \nabla_\delta p(\pi) \right]_{:,l} \left[ \nabla g(p(\pi)) \right]_{l,k} + \nabla_\theta p(\pi) \nabla \left[ \nabla g(p(\pi)) \right]_{l,k} \left[ \nabla_\delta p(\pi) \right]_{:,l}^T \right\}$$

6. $p(\pi) := h(y_i(\pi))$

   Similarly to the case of the above item, we have the following expression for the Jacobian of $p(\pi)$.

$$\nabla_\theta \left[ \nabla_\delta p(\pi) \right]_{:,l} = \nabla_\theta \left[ \nabla_\delta y_i(\pi) \nabla h(y_i(\pi)) \right]_{:,l}$$
$$= \nabla_\theta \left\{ \nabla_\delta y_i(\pi) \left[ \nabla h(y_i(\pi)) \right]_{:,l} \right\}$$
$$= \nabla_\theta \left\{ \sum_r \left[ \nabla_\delta y_i(\pi) \right]_{:,r} \left[ \nabla h(y_i(\pi)) \right]_{r,l} \right\}$$
$$= \sum_r \left\{ \nabla_\theta \left[ \nabla_\delta y_i(\pi) \right]_{:,r} \left[ \nabla h(y_i(\pi)) \right]_{r,l} + \nabla_\theta y_i(\pi) \nabla \left[ \nabla h(y_i(\pi)) \right]_{r,l} \left[ \nabla_\delta y_i(\pi) \right]_{:,r}^T \right\}$$

7. $y_i(\pi) := f(x_i; \theta) + \delta$

   For the Jacobian of $y_i(\pi)$ the following holds.

$$\nabla_\theta \left[ \nabla_\delta y_i(\pi) \right]_{:,r} = 0$$

To establish the Lipschitzness and boundedness of the expression in items 3-7, we follow the same steps and reasoning we used for the case of item 2. Ultimately, we can show that the function $\mathcal{L} \left( \pi; \{x_i\}_{i=1}^N \right)$ has bounded and Lipschitz Hessians/Jacobians.

**Part 3: Lipschitz continuity of $\nabla_\theta \widetilde{\mathcal{L}}((\theta, \phi), \delta)$, $\nabla_\phi \widetilde{\mathcal{L}}((\theta, \phi), \delta)$, $\nabla_\delta \widetilde{\mathcal{L}}((\theta, \phi), \delta)$**

Table 10: The outputs of the original (denoted as "OR") and the robustified model (denoted as "RO") on both the clean and the adversarial prompt of a given word. The words were obtained from the ImageNet-1K dataset.

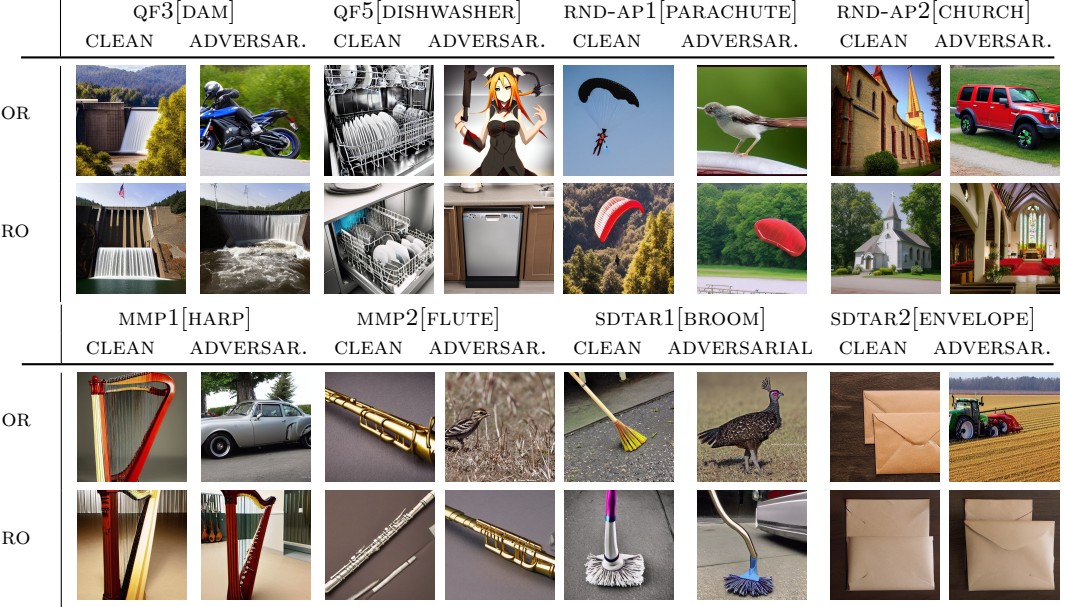

Finally, for $\nabla_\theta \widetilde{\mathcal{L}}((\theta, \phi), \delta)$ (and similarly for $\nabla_\phi \widetilde{\mathcal{L}}((\theta, \phi), \delta)$ and $\nabla_\delta \widetilde{\mathcal{L}}((\theta, \phi), \delta)$) and by using the results we developed in parts 1 and 2 (about the gradient boundedness and Lipschitz gradient continuity) we have

$$
\begin{aligned}
&\|\nabla_\theta \widetilde{\mathcal{L}}((\theta_1, \phi_1), \delta_1) - \nabla_\theta \widetilde{\mathcal{L}}((\theta_2, \phi_2), \delta_2)\| \\
&= \|\nabla_\theta \mathcal{L}((\theta_1, \phi_1), 0) + \nabla^2_{\theta\delta} \mathcal{L}((\theta_1, \phi_1), 0)\, \delta_1 - \nabla_\theta \mathcal{L}((\theta_2, \phi_2), 0) - \nabla^2_{\theta\delta} \mathcal{L}((\theta_2, \phi_2), 0)\, \delta_2\| \\
&\leq \|\nabla_\theta \mathcal{L}((\theta_1, \phi_1), 0) - \nabla_\theta \mathcal{L}((\theta_2, \phi_2), 0)\| \\
&+ \|\nabla^2_{\theta\delta} \mathcal{L}((\theta_1, \phi_1), 0)\, \delta_1 - \nabla^2_{\theta\delta} \mathcal{L}((\theta_1, \phi_1), 0)\, \delta_2 + \nabla^2_{\theta\delta} \mathcal{L}((\theta_1, \phi_1), 0)\, \delta_2 - \nabla^2_{\theta\delta} \mathcal{L}((\theta_2, \phi_2), 0)\, \delta_2\| \\
&\leq \|\nabla_\theta \mathcal{L}((\theta_1, \phi_1), 0) - \nabla_\theta \mathcal{L}((\theta_2, \phi_2), 0)\| \\
&+ \|\nabla^2_{\theta\delta} \mathcal{L}((\theta_1, \phi_1), 0)\|\|\delta_1 - \delta_2\| + \|\nabla^2_{\theta\delta} \mathcal{L}((\theta_1, \phi_1), 0) - \nabla^2_{\theta\delta} \mathcal{L}((\theta_2, \phi_2), 0)\|\|\delta_2\| \\
&\leq \hat{L}_2 \|(\theta_1, \phi_1, \delta_1) - (\theta_2, \phi_2, \delta_2)\|,
\end{aligned}
$$

for some proper Lipschitz constant $\hat{L}_2 > 0$.

The proof is complete.

$\square$

# E    Additional Results and Experiments

## E.1    Image Outputs Of The Original And The Robustified Model.

In Tables 10, 11 we present some image outputs from both the original and the robustified models across different attack configurations. The original model consistently fails to produce the correct image in every scenario. Conversely, the robustified model successfully generates the correct concept, i.e., the one that aligns with the clean prompt, under all the attacks considered in this work. Additionally, it is noteworthy that the robustified model can still produce the correct image when given a clean prompt as input.

Table 11: The outputs of the original (denoted as "OR") and the robustified model (denoted as "RO") on both clean and adversarial prompt across different attacks and words. The words were obtained from the CIFAR100 dataset.

Table 12: The FID score of the original and two of the robustified models, used in the experiments of table 2 (CIFAR100) and table 4 (ImageNet-1K). We note that the FID scores of the robustified models and the original one are close, which indicates that the former retain their ability to output accurate images when given as inputs the clean prompts.

| MODELS | ORIGINAL | ROBUST (CIFAR100) | ROBUST (IMAGENET-1K) |
|---|---|---|---|
| FID SCORE | 149 | 148 | 159 |

## E.2   FID Scores

In table 12 we report the FID score of the generated outputs of the original and two of the robustified models of our previous experiments, on clean prompts. Specifically, the robustified models used in tables 2 (CIFAR100) and 4 (ImageNet-1K). We stress that this set of clean prompts consists of all the words of the ImageNet dataset we provide in table 16, around 80 words, and not only the selected prompts we used in the rest of the experiments. The reason for performing the evaluation over a wider set is the need of having a large dataset for obtaining a reliable FID estimate; in fact, we generate 25 images per word over 80 words for the purpose of FID computation. Note that the FID scores attained by the robustified models are close to the score of the original one, which implies that the former maintain their ability to output accurate images on the clean prompts.

## E.3   Different Hyperparameter Configurations

In this section, we study the effect of using of different hyperparameter configurations. Specifically, we repeat the experimental process of Sec. 4.1, however this time we fix the values of certain hyperparameters that were previously tuned and we optimize the rest. We consider two additional hyperparameter configurations: 1) we set the number of ascent and descent steps to 1, i.e., $K = L = 1$; 2) we fix the weights assigned to the loss terms to 1, i.e., $\lambda_1 = \lambda_2 = 1$. In Table 13 we report the performance of the robustified model across four attacks on the three different configurations (i.e., the configuration of Sec 4.1 and the two new configurations)

We observe that the performance of the three different hyperparameter configurations on any specific attack (or the clean data) are relatively close, and there is no configuration that is clearly better than the others. We can, therefore, conclude that the method is robust to hyperparameter changes, as even if we fix some hyperparameters, the hyperparameter optimization process will find a satisfactory configuration with the remaining hyperparameters.

## E.4   Training and Model Parameters and Implementation Details

The majority of the model and training parameters are determined through a hyperparameter optimization process. These include the steps sizes $\mu_{att}, \mu_{def}$ and the number of iterations $K, L$ for the ascent and descent steps, respectively, the weights $\lambda_1, \lambda_2$ of the loss 5, the number of inference steps over which backpropagation

Table 13: The performance of the robustified model on different hyperparameter configurations. For our evaluation we use the classification accuracy ("CLASS") and the text-image similarity ("TEXT-IMAGE") score of the outputs; higher scores indicate better performance. The adversarially trained models were robustified against the QF5 attack and their performance is evaluated on three additional attacks (setting S2). The results are averaged over 5 different random instances (seeds) of the Stable Diffusion model; the average value and the full range of values attained over those different instances are reported.

| CONFIG. | MODEL | ROBUST | |
|---|---|---|---|
| | ATTACK/EVALUATION | CLASS | TEXT-IMAGE |
| MAIN EXP. | CLEAN | $1 \pm 0$ | $0.249 \pm 0.004$ |
| | QF5 | $0.690 \pm 0.060$ | $0.210 \pm 0.006$ |
| | QF3 | $0.660 \pm 0.073$ | $0.210 \pm 0.008$ |
| | MMP1 | $0.700 \pm 0.067$ | $0.224 \pm 0.004$ |
| | MMP2 | $0.463 \pm 0.030$ | $0.198 \pm 0.002$ |
| $K = L = 1$ | CLEAN | $0.989 \pm 0.056$ | $0.237 \pm 0.001$ |
| | QF5 | $0.575 \pm 0.091$ | $0.190 \pm 0.003$ |
| | QF3 | $0.675 \pm 0.108$ | $0.200 \pm 0.004$ |
| | MMP1 | $0.730 \pm 0.086$ | $0.216 \pm 0.003$ |
| | MMP2 | $0.531 \pm 0.053$ | $0.191 \pm 0.003$ |
| $\lambda_1 = \lambda_2 = 1$ | CLEAN | $0.978 \pm 0.044$ | $0.245 \pm 0.001$ |
| | QF5 | $0.722 \pm 0.077$ | $0.220 \pm 0.004$ |
| | QF3 | $0.706 \pm 0.061$ | $0.219 \pm 0.004$ |
| | MMP1 | $0.677 \pm 0.028$ | $0.216 \pm 0.001$ |
| | MMP2 | $0.531 \pm 0.064$ | $0.200 \pm 0.004$ |

is performed, etc. The rest of them are preselected. The fixed parameter values for the latter case and the range of parameter values for the former are provided in Table 14.

Table 14: The values or range of values of the model and training parameters used in the experiments.

| HYPERPARAMETER | HYPERPARAMETER VALUE/RANGE |
|---|---|
| INFERENCE STEPS | 16 OR 20 |
| IMAGE HEIGHT & WIDTH | 512 |
| $\mu_{att}$ | $[10^{-1}, 5]$ |
| $\mu_{def}$ | $[10^{-6}, 10^{-4}]$ |
| $\lambda_1$ | $[0.1, 20]$ |
| $\lambda_2$ | $[0.1, 20]$ |
| # BACKPROPAGATION STEPS | $\{6, 7, 8\}$ |
| # ADDITIONAL TOKENS (IN PERTURBATION) | $\{0, 1, 2\}$ |
| # DESCENT STEPS $K$ | $\{1, \ldots, 5\}$ |
| # ASCENT STEPS $L$ | $\{1, \ldots, 5\}$ |
| $\|\Delta\|_0$ | $[10^{-3}, 10^{-1}]$ |

In the experiments, we utilized the clip-vit-large-patch14 CLIP text and image encoders, the runwayml/stable-diffusion-v1-5 Stable Diffusion model, and the resnet-18 classifier that are available from Hugging Face (hug). The implementation of these models was carried out using the PyTorch and Hugging Face (Transformers, Diffusers) (hug) libraries. The robustification of each model was conducted on a single Nvidia A100 GPU with 40GB of memory.

### E.5 Word/Concept Lists

In our experiments, we utilize two distinct lists of words. From these lists, we select specific subsets to robustify and evaluate the Stable Diffusion model. These word lists are derived from a subset of the class labels in the CIFAR100 (Krizhevsky, 2009) and ImageNet-1K (Deng et al., 2009) datasets. More precisely, they were selected such that they correspond to high-level concepts (e.g., we favor concepts such as "dog" rather than "German shepherd") and their generated images can be unambiguously identified correctly by the classifier and by visual inspection. For example, there are class names for which the prompt "a photo of a [class name]" results in an image in the output (of the original model) that does not correspond correctly to the prompt. Such class names are excluded from our experimental evaluation to avoid misleading results. For example, in the case where the output of the robustified model is misclassified, we cannot state with certainty whether the source of failure is the adversarial training procedure, the T2I model, or the classifier.

Table 15: The CIFAR100 word list.

| CIFAR100 WORD LIST |
| --- |
| APPLE, AQUARIUM FISH, BEAR, BEAVER, BED, BEE, BEETLE, BICYCLE, BOTTLE, BOWL, BRIDGE, BUS, BUTTERFLY, CAMEL, CASTLE, CATTLE, CHAIR, CHIMPANZEE, CLOCK, COCKROACH, COUCH, CROCODILE, CUP, DINOSAUR, ELEPHANT, FOX, HAMSTER, KANGAROO, KEYBOARD, LAWN MOWER, LEOPARD, LION, LIZARD, LOBSTER, MOTORCYCLE, MOUSE, MUSHROOM, ORANGE, OTTER, PICKUP TRUCK, PLATE, PORCUPINE, RABBIT, RAY, ROCKET, SEA, SHARK, SKUNK, SNAIL, SNAKE, SPIDER, SQUIRREL, STREETCAR, SWEET PEPPER, TABLE, TANK, TELEPHONE, TELEVISION, TIGER, TRACTOR, TRAIN, TURTLE, WARDROBE, WHALE, WOLF |

Table 16: The ImageNet-1K word list.

| IMAGENET-1K WORD LIST |
| --- |
| AIRCRAFT CARRIER, AMBULANCE, BALLOON, BANANA, BASKETBALL, BINOCULARS, BOATHOUSE, BOOKCASE, BROOM, BUBBLE, CANDLE, CD PLAYER, CELLULAR TELEPHONE, CHAIN SAW, CHURCH, CINEMA, CLIFF, CONTAINER SHIP, DAM, DISHWASHER, DOORMAT, DRILLING PLATFORM, DRUM, ELECTRIC GUITAR, ENVELOPE, ESPRESSO, ESPRESSO MAKER, FLUTE, FOUNTAIN, FRYING PAN, GARBAGE TRUCK, GEYSER, GOLFCART, GONDOLA, GRAND PIANO, GREENHOUSE, HARP, HATCHET, HOURGLASS, ICE CREAM, IPOD, JOYSTICK, LAPTOP, LEMON, LIBRARY, LIFEBOAT, LIGHTER, LOUDSPEAKER, MISSILE, MONASTERY, MOUSE, OSCILLOSCOPE, PAINTBRUSH, PALACE, PARACHUTE, PENCIL BOX, PETRI DISH, PHOTOCOPIER, PINEAPPLE, PIZZA, PLANE, PRINTER, PRISON, RADIATOR, RADIO, RADIO TELESCOPE, RED WINE, RESTAURANT, RIFLE, SOCCER BALL, SOCK, SPACE SHUTTLE, SPEEDBOAT, STETHOSCOPE, STOVE, STRAWBERRY, SUBMARINE, SUIT, SUNGLASSES, SYRINGE, TENNIS BALL, THRONE, TORCH, TOW TRUCK, UMBRELLA, VALLEY, VIOLIN, VOLCANO, WALLET, WATER BOTTLE, WATER TOWER |

