# OpenReview forum: "An Adversarial Training Approach to Robustify Stable Diffusion Systems Against Prompting Attacks"
_TMLR — Rejected by TMLR_

### Review · Reviewer_iYqn · 2025-02-03

**Summary Of Contributions:**

The paper proposes to apply adversarial training in text-image diffusion models. Specifically, it studies where the perturbation needs to be added in the diffusion system since the text prompt. They found that the embedding after the text encoder would be a good choice. They then design two loss functions to make the adversarial training. The first loss is based on the classification accuracy of the generated image and the second loss is on the similarity score by projecting text and image to their respective encoders in the CLIP model. Two optimization methods are discussed to solve the problem. The experimental results show the proposed method can achieve better robustness on several adversarial attacks.

**Audience:**

Yes

**Broader Impact Concerns:**

No ethical concerns.

**Claims And Evidence:**

No

**Requested Changes:**

Please refer to the weaknesses. To sum,
1. Experiments with more datasets and models need to be added.
2. Ablation studies on hyperparameters and some basic baselines needed to be considered.

**Strengths And Weaknesses:**

Pros:
1. The paper is easy to follow and well-written, except for some minor typos.

Cons:
1. Although the paper claims it is the first paper to apply adversarial training to a diffusion model, the proposed method is not very different from normal adversarial training in the classification model, which limits its novelty. The proposed method even uses a classifier in the end to make the proposed method look no different than normal adversarial training. The unique challenge introduced in the text space is already being discussed a lot in text adversarial examples.
2. The usage of a classifier as the adversarial training objective is problematic. It will greatly restrict the perturbation space to only certain classes that can be classified and they have to be given in the first place. I don't think the setting is reasonable under the diffusion setting, where the paper only uses a couple of labels in their experiments. To sum up, it is not a good idea to treat the generative model just as a classifier.
3. There are no other baselines and ablation studies on the hyperparameter in the paper, making the paper weak in the experiments.  Only one dataset and one model are used in the paper, which is far from sufficient.

Minor:
1. Two attacks method in Figure 4 is not introduced until the experiments.
2.  $L_3$ in section 3.3 should be $L_{img}$.
3. Quotation issues in section 3.1.

---

> ### Author Response · Authors · 2025-03-04
> **Response**
>
> >1.Although the paper claims it is the first paper to apply adversarial training to a diffusion model, the proposed method is not very different from normal adversarial training in the classification model, which limits its novelty. The proposed method even uses a classifier in the end to make the proposed method look no different than normal adversarial training.The unique challenge introduced in the text space is already being discussed a lot in text adversarial examples.
>
> We thank the reviewer for the comment, however we do not fully agree with the assessment. We believe that the proposed method is sufficiently novel, extending the adversarial training (AT) method from classification to diffusion models requires addressing a number of challenging issues, such as identifing the suitable loss function and the trainable components. Moreover, the utility of the classifier in our method is different compared to its utility in classical AT. Specifically, in our case the classifier is **not** the final model that is being robustified; it is instead used (as one of a few measures) to evaluate both the quality of the output and its alignement with the input text. Therefore having a classifier does not make our method similar to classical AT.  Finally, while issues related to the text space have been discussed in other works, they do not diminish the novelty of our work, as our approach leverages some very specific structure of T2I systems (unlike these other works). We explain our reasoning in detail below.
>
> ---
>
> First, while the proposed defense adopts the concept of AT from image classification and transfers it to T2I systems, it is **not** a straightforward extension of AT. The setting of T2I systems differs significantly from that of classification problems, and consequently, applying AT involves unique challenges and design considerations. Extensive experiments were conducted to identify the correct approach. The considerations are the following:
> - (1) How to Robustify?
> To begin with, it is not clear how to formulate an AT problem that robustifies T2I systems. Unlike classical AT where a simple classification loss suffices to capture the effectiveness of an adversarial perturbation, T2I systems are intrinsically multimodal. Therefore, the design of loss functions needs to take into consideration **both** text and image qualities. Second, it is not clear what is the mechanism that one should choose to robustify the T2I system. Again, unlike the traditional AT which directly adds perturbation to the image domain, which lies in a continuous space, in T2I system the input is in the text space. Directly applying existing AT methods will result in a discrete optimization problem which can be intractable to solve. Therefore, it is imperative to explore other options, such as constructing an embedding space or using one that is already available.
> - (2) What to Robustify?
> Stable Diffusion consists of several unique components with trainable parameters, e.g., a text encoder, a UNet. This is significantly different as compared to the majority of existing AT methods, which typically operate on CNN-based neural networks. To determine which subset of trainable parameters to optimize, one needs to investigate and understand the tradeoff between various performance metrics, such as robustness and computational efficiency. **Our investigation indicates that**, rather than tuning the prameters of all the model's components, we only need to focus on the text encoder and the VAE decoder while keeping the parameters of the UNet frozen.
> - (3) How to Implement Efficiently?
> T2I models are computationally more demanding as compared to the traditional CNN-based neural networks, with even a forward pass incurring significant runtime and memory costs, not to mention the even more computationally expensive backpropagation steps needed during the model training process. Therefore, special care must be taken to design practical and computationally efficient AT methods for T2I systems. **Our investigation indicates** that we can improve the efficiency of the AT method by storing the computational graph for a number of steps fewer than the number of inference steps and by keeping the number of iterations small, without compromising the model's robustness.
>
> We note that the discussion above is already included in the text, specifically in the introduction section.

---

> ### Author Response · Authors · 2025-03-04
> **Response (2)**
>
> [Continuing the answer from above.]
>
> ---
>
> Second, the utility of the classifier in our method differs significantly from its utility in classical AT. In classical AT the classifier is the model that is robustified against adversarial perturbations. On the contrary, in our setting the model that is robustified against adversarial attacks is the Stable Diffusion T2I system, which outputs images, and is not a classifier.
>
> More specifically, in classical AT, the classifier is only needed to obtain the label of the perturbed image to determine whether the perturbation is adversarial (i.e., if it results in an image with a different class label) or not. Therefore, the classifier's native loss function suffices to capture the AT objective, i.e.,
> $$\mathcal{L} = \ell_{cls} (\bar{x},u(\theta,\delta)),$$
> where $u$ is the classifier, $\theta$ is the classifier's parameters, $\delta$ is the perturbation (which is added in the image) and $\bar{x}$ is the ground truth label.
>
> On the other hand, in T2I systems, due to their multimodal nature, we potentially need to evaluate multiple aspects of the image output, including its quality, its correlation with the text input, and its distance from a ground truth image (i.e., image generated by a clean prompt). As this is a novel application domain for AT, it is not  known *a priori* how to evaluate the T2I system's output and there are potentially many different valid ways to do so. In our case we selected three different loss terms to evaluate the quality of the output images: classification loss, text-image similarity (clip score), and output sensitivity to capture these aspects. Through careful experimentation we established that two of those loss terms suffice to provide good performance, i.e.,
> $$\mathcal{L} \left( (\theta,\phi), \delta; x \right) =
>     \lambda_{1} \cdot \ell_{\text{sim}} \left( \widehat{f}(x), w(\theta,\phi,\delta) \right)
>     + \lambda_{2} \cdot \ell_{\text{cls}}\left(\bar{x},u(\theta,\phi,\delta) \right),$$
> where $(\theta,\phi)$ are the T2I model parameters, $u$ is the classifier, $\delta$ is the perturbation (which is added in the text encoder embedding space), $\bar{x}$ is the ground truth label, $w$ is the image encoder and $\widehat{f}$ is the text encoder.
>
> Overall, in our setting, the classifier is just one of the metrics we use to evaluate the output. Its utility is twofold, as it is used to assess both the quality of the output and its alignment with the input text.
>
> ---
>
> Finally, as the reviewer mentioned, the challenges associated with working in the text space have already been discussed in the literature of text adversarial examples (e.g., [R1, R2]). We agree that **the challenge is not new**, however, the **solutions** to this challenges can be quite different for different systems. Indeed, this literature has been focusing on models different from T2I, such as RNNs, CNNs, and LLMs. While it is natural that there are similarities between the challenges encountered in T2I systems and those in other models, as they have the same type of input (i.e., text), the structural differences of these models demands that model-specific solutions have to be developed.
>
> In our work, our solution for mitigating the challenges of working in the discrete text space **leverages the particular structure of the Stable Diffusion T2I model**. Specifically, we avoid operating in the text space by utilizing the continuous text encoder space provided within Stable Diffusion. It is in this space that we perturb the prompts and identify the relations between clean and adversarial prompts; please see Section 3.1 and Figure 4 for more details. As this specific structure is unique to Stable Diffusion, it makes our approach distinct from the discussions in the aforementioned literature. As a result, the fact that the challenges we encounter might not be novel does not diminish the overall novelty of our work.
>
> [R1] Wang, Wenqi, et al. "Towards a robust deep neural network in texts: A survey." _arXiv preprint arXiv:1902.07285_ (2019).
>
> [R2] Wang, Xuezhi, Haohan Wang, and Diyi Yang. "Measure and improve robustness in NLP models: A survey." _arXiv preprint arXiv:2112.08313_ (2021).

---

> ### Author Response · Authors · 2025-03-04
> **Response (3)**
>
> > 2.The usage of a classifier as the adversarial training objective is problematic. It will greatly restrict the perturbation space to only certain classes that can be classified and they have to be given in the first place. I don't think the setting is reasonable under the diffusion setting, where the paper only uses a couple of labels in their experiments. To sum up, it is not a good idea to treat the generative model just as a classifier.
>
> Thanks for the comment. However, it appears to us that there is a misunderstanding of our approach. Specifically, we note that the classifier does **not** restrict the perturbation space to correspond only to certain classes. Therefore, we do **not** treat the generative model just as a classifier.
>
> To begin with, let us clarify our understanding of the term  "restricting the perturbation space". We think the reveiwer meant the following: One consider only those perturbations in the text encoder space, for which the corresponding perturbed prompt (formulated by adding the perturbation to the original prompt) results in images belonging exclusively to certain classes, specifically those supported by the classifier.
>
> We stress that **this type of restriction does not apply in our case**. This is because, we are only interested in whether the generated image belongs to the class described by the original (unaltered) prompt (which we can always check as we are considering prompts of the form "a photo of a [CLASS LABEL]"). If it does not belong to the class of the original prompt, then the perturbed prompt is "adversarial", regardless of the class of the generated image. Therefore, the actual class of the "adversarial" prompt is not relevant and thus there is no real problem if the generated image cannot be classified under one of the available classes. Overall, this means that **any perturbation is valid**, regardless if the respective generated image can be classified under the given classifier.
>
> Finally, we would also like to stress here that the classifier is only one of the two evaluation mechanisms we are using in our method. We are also using the CLIP score to evaluate the alignment between the original prompt and the generated image. As a result, we are not solely relying on the classifier for the constructing and evaluating adversarial perturbations.
>
> > 3.There are no other baselines and ablation studies on the hyperparameter in the paper, making the paper weak in the experiments. Only one dataset and one model are used in the paper, which is far from sufficient.
>
> Thanks for the feedback. We note that instances of the experiments proposed by the reviewer are already present in our work. In addition, we added an ablation study on the hyperparameters in the revised paper.
>
> To begin with, the appendix already included experiments on different datasets and models. Specifically, in section 'E.2 Different dataset,' we robustify the model against various labels from the ImageNet dataset (whereas the main text uses labels from CIFAR100). In section 'E.4 Different version of the Stable Diffusion model', we conduct experiments using the Stable Diffusion v1.4 model instead of the v1.5 model used in the main text. In both cases, the results show that our method performs well under different settings (e.g., models, datasets). In the revised version we moved these experiments from Appendix E to the main text, specifically after the end of the section "4.1.1 Results".
>
> Furthermore, in the appendix, we conducted several experiments that can be considered ablation studies, as they examine the impact of specific components (and their hyperparameters) on the performance of the robustified model. More precisely, in section "B Evaluation of the UNet", we examined the effect of updating the UNet parameters, which remain fixed in the rest of our experiments. Additionally, in section "C Evaluation of the Output Sensitivity Loss Term", we investigated the impact of incorporating the output sensitivity term (defined in eq. (4)) into our loss function. Finally, in the revised paper (Appendix E.3) we added an experiment where we study the effect of using different hyperparameter configurations. Specifically, we fix the values of certain hyperparameters that were previously tuned and we optimize the rest.
>
> Finally, while other studies develop defense techniques against specific forms of misuse, none share the same goal as our approach. Therefore, to our knowledge, there are no other suitable baselines.

---

> ### Author Response · Authors · 2025-03-04
> **Response (4)**
>
> >1.Two attacks method in Figure 4 is not introduced until the experiments.
>
> Thanks for the comment. We will add a reference to the caption.
>
> > 2.L3 in section 3.3 should be Limg.
>
> Thanks for noticing. We will fix this issue.
>
> > 3.Quotation issues in section 3.1.
>
> Thanks for noticing. We will fix this issue.
>
> Requested Changes:
> > 1.Experiments with more datasets and models need to be added.
>
> Please refer to the 3rd point under the 'Cons' section above for our response.
>
> > 2.Ablation studies on hyperparameters and some basic baselines needed to be considered.
>
> Please refer to the 3rd point under the 'Cons' section above for our response.

---

### Review · Reviewer_QMMP · 2025-02-11

**Summary Of Contributions:**

In this work, the authors propose a new method to robustify stable diffusion models from prompting attacks. The idea is baed on adversarial training from the image classification model to solve a min-max optimization problem. Experimental results showed that model trained using the proposed objective is able to improve robustness against different prompting attacks.

**Audience:**

Yes

**Claims And Evidence:**

No

**Requested Changes:**

- I strongly suggest rewriting the objective to be $$\min_{\theta,\phi}\max_{\delta}\sum_{i=1}^nL_i, L_i=-\lambda_1l_{cls}-\lambda_2l_{sim}$$
- Explain / revisit the correctness of the objective, as I pointed out in the second weakness.
- Explain motivation w.r.t the third weakness.

**Strengths And Weaknesses:**

Strengths:
- The study of robustness against prompt based attack is important and the application of adversarial training is interesting.
- Empirical results seem strong against different attacks.

Weaknesses:
- The objective (1) is confusing. Different from the adversarial training objective, the proposed method aims to solve a min max optimization problem which find the model weight $w$ that *maximizes* the loss instead of *minimizes*. It only makes sense until page 7 where the author defines the loss 2 and 3, where the author explicitly explain the adversary's goal is to make the $l_{cls}$ and $l_{sim}$ large. I strongly recommend either removing objective 1 or push loss 2 and 3 up in order to avoid confusion.
- I don't understand the motivation of solving $\min_\delta$. I thought the optimization goal is to find the best model under the worst adversarial example. Shouldn't the worst adversarial example correspond to the sample with max noise scale with the singleton $\Delta$. Formulated in its current way, different $\Delta$ will give the same results. Hence, I'm concerned about the correctness of the optimization objective.
- Mismatch between the proposed objective / algorithm and the experimental results. Adversarial training in image classification model could be done this way because the threat model itself is usually defined as $\ell_2$ or $\ell_\infty$ ball. It does not seem that the attacks evaluated here assumes a $\ell_2$ or $\ell_\infty$ ball.
- By text image similarity score, do you mean the CLIP score?
- Missing evaluations of white box adaptive attacks to the proposed defense.

---

> ### Author Response · Authors · 2025-03-04
> **Response**
>
> > *The objective (1) is confusing. Different from the adversarial training objective, the proposed method aims to solve a min max optimization problem which find the model weight $w$ that maximizes the loss instead of minimizes. It only makes sense until page 7 where the author defines the loss 2 and 3, where the author explicitly explain the adversary's goal is to make the lcls and lsim large. I strongly recommend either removing objective 1 or push loss 2 and 3 up in order to avoid confusion.*
>
> To clarify, let us redefine the loss function $\mathcal{L}$ as $\mathcal{L}=\sum_{i=1}^n -\lambda_1l_{cls}-\lambda_2l_{sim}$ as the reviewer suggests below (in the "requested changes"). Then, the objective (1) will be rewritten in the more familiar min-max form, similar to how adversarial training is typically defined, i.e., $\min_{\theta,\phi}\max_{\delta}\mathcal{L}$ . In this case, there will be no need to define losses 2 and 3 earlier, which is not possible anyway, as introducing these losses requires lengthy explanations that are not suitable at the beginning of Section 3. The above redefinition as well as all the other changes that follow from it (e.g., change in the algorithms's steps) are highlighted with blue in the revised text.

---

> ### Author Response · Authors · 2025-03-04
> **Response(2)**
>
> > I don't understand the motivation of solving $min_{δ}$. I thought the optimization goal is to find the best model under the worst adversarial example. Shouldn't the worst adversarial example correspond to the sample with max noise scale with the singleton Δ. Formulated in its current way, different Δ will give the same results. Hence, I'm concerned about the correctness of the optimization objective.
>
> Thanks for the feedback. First, we agree with the reviewer that optimization goal is to find the best model under the worst adversarial example. The worst adversarial example is the most effective/damaging one according to some effectiveness measure and under certain restrictions (on the ways we can modify the original prompt to get the adversarial one). However, we note that there is no universally accepted method to identify and evaluate such adversarial examples and there are potentially many different valid ways to do so. In our work we propose one such way. Specifically, we construct a loss function that quantifies how effective an adversarial prompt is, allowing it to capture the worst adversarial example in a certain sense and within a given budget (with respect to the magnitude of the perturbation). This loss function consists of two terms:
> - **Text-Image Similarity (CLIP Score).** This measure evaluates the deviation between the image generated by the perturbed prompt and the clean text. It is defined as the correlation between the embeddings of a text and an image, and these embeddings are obtained by projecting text and image to their respective encoders in the CLIP model. Mathematically, we can write $\ell_{sim}(u,v) = u^{T}v / \|u\|\|v\|$ where $u,v$ are the text and image embedding vectors, respectively. When the attack is effective, $\ell_{sim}$ is small. Finally, we note that this measure has also been used in the literature in the same manner, i.e.  to quantify the effectiveness of attacks (e.g., in the QF and MMP attacks).
> - **Classification Loss.** This measure evaluates the quality of the perturbed output image. It is defined as the the logarithm of the probability that the generated image belongs to the ground truth class; the ground truth is the label of the image generated by the clean prompt. Mathematically: $\ell_{\text{cls}}(u,v) = \log(u^{T}v)$ where the vector $u$ denotes the softmax output of the classifier and $v$ is a vector of all zeros except at the index corresponding to the ground truth class. An effective attack will make  $\ell_{\text{cls}}$ small, as the probability that the images generated by it and the clean prompt belong to the same class is small.
>
> Then, the two terms are combined to formulate the objective $\mathcal{L}((\theta,\phi),\delta)=\sum_{i=1}^n -\lambda_1l_{cls}((\theta,\phi),\delta)-\lambda_2l_{sim}((\theta,\phi),\delta)$. Based on this definition the worst adversarial prompt will correspond to the perturbation $\delta$ that results to the highest loss $\mathcal{L}$.
>
> Finally, we note that different constraints sets $\Delta$ (e.g.,  $\ell_{2}$ or $\ell_{\infty}$ balls, different radii) will not necessarily give the same results. This is because the inner max problem is a non-concave one and as a result the set of global minima of the objective $\mathcal{L}$ can change as we change the constraints.
>
> To make things more clear let us provide a simple example which is a special case of our formulation. Consider the objective function $f(x,y)=x^{2}-y^{2}$ and the optimization problem $\min_{x\in X}\max_{y\in Y} f(x,y)$, where $X=\{-1 \leq x \leq 1\}$ and $Y=\{a \leq x \leq b\}$. This problem is actually easier than our own as the objective is strongly-convex in $x$ and strongly-concave in $y$. The solution of this optimization problem relies on the parameters $a,b$ of the set $Y$.  For instance, if $a=-1,b=1$, the  solution of the inner max problem is $y^*=0$ and that of the outer problem is $x^*=\arg\min_{x\in X} f(x,0)=0$, hence the overall solution is $(x^*,y^*)=(0,0)$. On the other hand, if $a=1,b=2$,  the  solution of the inner max problem is $y^*=1$ and that of the outer problem is $x^*=\arg\min_{x\in X} f(x,1)=0$, hence the overall solution is $(x^*,y^*)=(0,1)$. Therefore, we clearly see how the change in the constraints set has an effect in the problem's solution.

---

> ### Author Response · Authors · 2025-03-04
> **Response (3)**
>
> > Mismatch between the proposed objective/algorithm and the experimental results. Adversarial training in image classification model could be done this way because the threat model itself is usually defined as $\ell_{2}$ or $\ell_{\infty}$ ball. It does not seem that the attacks evaluated here assumes a $\ell_{2}$ or $\ell_{\infty}$ ball.
>
> Thanks for the comment. Similar to adversarial training (AT) for image classifiers we make a choice about the underlying threat model. This choice is reflected in the optimization formulation and algorithm and therefore there is no mismatch between the "objective/algorithm and the experimental results". However, differently than classical AT it is not immediately clear what the threat model is as the adversarial attacks are developed in the text space. As a result, part of our work is devoted to identifying a suitable way of modelling the adversarial prompts. We provide more details below.
>
> To begin with, we note that adversarial prompt attacks are originally developed on the text space. However, as the text space is discrete, optimizing perturbations in that space makes the problem combinatorial and thus challenging. Therefore, we choose to work instead on the continuous embedding space of Stable Diffusion's text encoder. It is in this continuous space that we seek to identify a suitable threat model for the perturbation. To this end we construct t-SNE plots of the embeddings of the adversarial prompts in the text encoder space; see figure 4(b) (pg. 6). We note that in the text encoder space the vectors representing adversarial prompts cluster nicely around those of clean ones. This suggests a **key observation**, that a small perturbation around clean vectors can be used to simulate adversarial attacks in the prompt space. Therefore, our threat model is one in which the perturbation is defined in a continuous space and its magnitude is small, and thus it can be described in a manner similar to classical AT, i.e., by considering as a constraint set for the perturbation $\delta$ a small  $\ell_{2}$ or $\ell_{\infty}$ ball.
>
> > By text image similarity score, do you mean the CLIP score?
>
> Yes, we use the term "text-image similarity" throughout the text to refer to CLIP score. We will add a comment in the revised text to clarify that.
>
> > Missing evaluations of white box adaptive attacks to the proposed defense.
>
> Thanks for the comment. However, to our knowledge there are no suitable white-box adversarial attacks for T2I systems.
>
> > I strongly suggest rewriting the objective to be $$\min_{\theta,\phi}\max_{\delta}\sum_{i=1}^nL_i, L_i=-\lambda_1l_{cls}-\lambda_2l_{sim}$$
>
> Thanks for the suggestion. We plan to follow the reviewer's suggestion in order to make our presentation more clear. Please also see the 1st point under the 'Weaknesses' section above.
>
> > Explain / revisit the correctness of the objective, as I pointed out in the second weakness.
>
> Thanks for the suggestion. In the paper, there is significant discussion about the structure of the objective and an explanation about how it quantifies the effectiveness of the adversarial prompts. Please see Sections 3.3 and 3.4. Also, for more details about the explanation please refer to the 2nd point under the 'Weaknesses' section above for our response.
>
> > Explain motivation w.r.t the third weakness.
>
> Thanks for the suggestion. In the paper, there is significant discussion about what constitutes a suitable choice for the continuous embedding space and the relationship between the clean and adversarial prompts within it, e.g., see Section 3.1 and Figures 4,6,7. Nonetheless, at the end of Section 3.1, in the revised paper, we added a comment to highlight how these observations are reflected in the optimization formulation. Please also refer to the 3rd point under the 'Weaknesses' section above for more details.

---

### Review · Reviewer_Embr · 2025-02-18

**Summary Of Contributions:**

This paper introduces an adversarial training (AT) method, MAT-SD, to enhance the robustness of Stable Diffusion models against specific types of prompting attacks. The core objective is to ensure that generated images align with intended concepts even when prompts are adversarially manipulated. The authors highlight the novelty of their approach in applying AT to T2I systems, addressing the unique challenges posed by the multimodal nature of T2I and the textual input space.

**Audience:**

Yes

**Claims And Evidence:**

Yes

**Requested Changes:**

In addition to addressing the weaknesses mentioned previously, I have the following writing comments (request for changes in writing):

- Add details to motivation: While the paper distinguishes its goal from existing work on harmful output prevention, it would be helpful to strengthen the rationale behind its specific objective. Explaining the broader implications and potential impacts of ensuring prompt alignment, regardless of harmfulness, would be valuable.


- The contribution of "MAT-SD as an implementation-friendly version of the HiBSA algorithm" requires more context. Explaining the significance of this adaptation and its advantages over the original HiBSA algorithm would clarify its contribution.


- Conclusion, limitations, and future work:
-- The conclusion should concisely summarize the limitations of the proposed method to provide a balanced overview.
-- Consider adding insights on the method's effectiveness and potential advantages/drawbacks when applied to other datasets. This would be a valuable addition to the results or following sections.

**Strengths And Weaknesses:**

Strengths:

- Novelty: The paper tackles a critical issue in T2I systems: their vulnerability to prompt manipulation. This is timely and addresses a growing concern in the field. The paper sets itself well in the current literature on adversarial attacks and associated mitigation methods in T2I.
- Clear methodology reasoning and description: The authors provide a detailed explanation of their AT approach, including the implementation considerations and the specific attack generation strategy. The clarity in their explanations of the methods and reasoning in the paper, especially in Sections 3.1 and the 3.5, and the outline of the unique challenges in the introduction section is particularly commendable. The introduction effectively outlines the key challenges in applying AT to T2I systems, such as formulating appropriate loss functions and choosing effective robustification mechanisms.
- Considerations of limitations of method proposed: The discussion of the proposed method's limitations in handling RND-AP type attacks demonstrates a thorough approach to evaluation of the methods proposed.

Weaknesses:

- Lack of specificity in defining the set of adversarial attacks covered by AT:
The abstract and introduction could benefit from a more precise definition of the specific class of adversarial attacks targeted by AT. Currently, terms like "adversarial modifications" and "this type of misuse" are vague. Specifying whether the focus is on character-level perturbations, semantic alterations, or other attack vectors would significantly enhance clarity and help scope the paper early on. Please consider clarifying the statement "certain classes of prompting attacks" in the abstract, providing more exact and accurate wording.

- Generalizability and Robustness Evaluation:
In section 4.1 the paper mentions “We will show below that not all of the prompts of RND-AP lie close to their corresponding clean ones. However, we opt to keep the RND-AP attack as it will allow us to showcase the limitations of our method.”
-- It is commendable that the paper shows its limitations in generalizing to different attack strategies based on the distance in the text embedding space. I have some follow up questions on the generalizability of the method:
-- Is it possible to reconstruct prompts that would lead to the text embedding vectors generated by the attack strategy?
-- What are some other methods like RND-AP that do not conform to the attack strategy chosen in the paper? More generally, it would be good to understand what features of an attack strategy are necessary or sufficient to have, for that attack strategy to provide similar outcomes in the text embedding space as those shown in the paper
-- Defining a class of adversarial attacks that the method generalizes to would also help clarify the message and scope of this work in the abstract and introduction

- Lack of consideration of negative effects of adversarial training on model performance
To understand the usefulness of the attack mitigation strategy proposed in this work, it would be important to add experiments to the paper to show that the robustified model does not degrade in performance by overemphasising on the concept when it is a non-adversarial prompts or innocuous prompts such as “bicycle thief”, or “bicycle CF SLX 8” (which is the model name of a specific bicycle).
More generally, it is important to consider potential negative effects of the method on model performance.

---

> ### Author Response · Authors · 2025-03-04
> **Response**
>
> > Lack of specificity in defining the set of adversarial attacks covered by AT: The abstract and introduction could benefit from a more precise definition of the specific class of adversarial attacks targeted by AT. Currently, terms like "adversarial modifications" and "this type of misuse" are vague. Specifying whether the focus is on character-level perturbations, semantic alterations, or other attack vectors would significantly enhance clarity and help scope the paper early on. Please consider clarifying the statement "certain classes of prompting attacks" in the abstract, providing more exact and accurate wording.
>
> Thanks for the comment. We edited the Introduction section to specify more accurately the type of attacks against which our adversarial training (AT) is developed. Specifically, we made clear that our AT method is developed to work against certain classes of prompting attacks where the embeddings of the clean and adversarial prompts are close in a certain continuous text embedding space. Please see the highlighted text in Sections "1 Introduction" and "1.2 Contributions" in the revised paper. Moreover, we clarified the statement "certain classes of prompting attacks" in the abstract. Specifically, we expanded the the final sentence as follows: "Finally, through several experiments, we demonstrate that the proposed method enhances the model's robustness against classes of prompting attacks where the embeddings of the clean and adversarial prompts are close in a certain continuous text embedding space."
>
> > Is it possible to reconstruct prompts that would lead to the text embedding vectors generated by the attack strategy?
>
> Yes, it is possible to reconstruct the prompts corresponding to the text embedding vectors. However, implementing a decoder that reconstructs the prompts is not trivial, and there is no practical utility in our method in doing so. Therefore, we do not pursue it.
>
> To be more precise, reconstructing a prompt requires designing a decoder that transforms an embedding to the corresponding text prompt. The decoding process requires projecting all the words of a dictionary in the text encoder's embedding space and finding the embeddings closer to the given projections. This involves solving an optimization problem for every embedding we would like to reconstruct. As a result, the reconstruction process is not trivial. Moreover, the reason we resort to the text encoder's embedding space is to avoid using the text space. This is because the text space is discrete and that makes the perturbation generation problem a combinatorial optimization problem, which is typically difficult to solve. Essentially we transfer our perturbation generation problem from the discrete text space to a continuous embedding space where we can solve it more easily. Therefore, returning back to the text space has no utility in our method.
>
> > What are some other methods like RND-AP that do not conform to the attack strategy chosen in the paper?
>
> In general, we observed that typo-like attacks do not conform to the attack strategy chosen in this paper. With typo-like attacks we are referring to prompt modifications such as adding or removing letters from the prompt, substituting letters and flipping the order of two consecutive letters. In fact, we studied the embeddings of such types of modifications and noticed patterns similar to the ones of RND-AP.
>
> > More generally, it would be good to understand what features of an attack strategy are necessary or sufficient to have, for that attack strategy to provide similar outcomes in the text embedding space as those shown in the paper
>
> To begin with, we note that our understanding of the class of attacks our method is designed for is based mostly on empirical observations. Therefore, it is not possible to conclusively identify a set of features that an attack must have to behave in the text embedding space in the desired manner (i.e., similarly to the QF, MMP, and SDTAR attacks). However, in practice, we observed that the attacks conforming to the attack class described in our work are those that append a few tokens to the clean prompt.
>
> > Defining a class of adversarial attacks that the method generalizes to would also help clarify the message and scope of this work in the abstract and introduction
>
> As we mentioned in our response to the first weakness above, we will provide some clarifications about the class of adversarial attacks we are defending against  in the abstract and introduction.

---

> ### Author Response · Authors · 2025-03-04
> **Response (2)**
>
> > Lack of consideration of negative effects of adversarial training on model performance To understand the usefulness of the attack mitigation strategy proposed in this work, it would be important to add experiments to the paper to show that the robustified model does not degrade in performance by overemphasising on the concept when it is a non-adversarial prompts or innocuous prompts such as “bicycle thief”, or “bicycle CF SLX 8” (which is the model name of a specific bicycle). More generally, it is important to consider potential negative effects of the method on model performance.
>
> Thanks for the comment. In our experiments we include results that quantify the effects of adversarial training on non-adversarial prompts. Specifically, we present the following three types of results:
> - **Performance of the robustified model on clean prompts.** In our main results (Tables 1-5), along with evaluating the performance of the robustified model on adversarial prompts, we also assess its performance on clean prompts, e.g., on prompts such as "a photo of a bicycle". This evaluation includes both the classification accuracy and the text-image similarity (clip score). Overall, we observe that the robustified model retains its ability to generate the correct image when provided with a clean prompt.
> - **FID score of robustified model on clean prompts.** In the appendix (Table 12), we evaluate the FID score of the robustified model by examining the outputs of the model on about 80 clean prompts. We note that the FID scores attained by the robustified models are close to the FID score of the original one, which implies that the former models maintain their ability to output accurate images.
> - **Example image output.** In the appendix (Tables 10,11) we present the image outputs of the original (i.e., not adversarially trained) and the robustified model on a number of clean and adversarial prompts. We note that the robustifeid model manages to generate pictures whose quality is similar to the quality of the original  model.
>
> > *Add details to motivation: While the paper distinguishes its goal from existing work on harmful output prevention, it would be helpful to strengthen the rationale behind its specific objective. Explaining the broader implications and potential impacts of ensuring prompt alignment, regardless of harmfulness, would be valuable.*
>
> Thanks for the comment. We will provide more details about our motivation in the revised version, specifically in Section "2.2 Objective and Application Scenarios".
>
> Indeed, ensuring the robustness of the model against prompt manipulations is important, regardless of the harmfulness (harmful in the sense of depicting typical harmful concepts, such as violence, nudity, or suicide) of the output. More precisely, the generation of a mismatched output, even if not harmful, can significantly degrade the user's experience. Moreover, the use of targeted attacks, which modify a clean prompt in a specific manner, can result in the generation of unauthorized content. For instance, such content could include the depiction of a political figure in an unwanted background or the modification of a company's logo or of a national symbol. Overall, all these scenarios can result in damage to the model's and the model owner's reputation.

---

> ### Author Response · Authors · 2025-03-04
> **Response (3)**
>
> > The contribution of "MAT-SD as an implementation-friendly version of the HiBSA algorithm" requires more context. Explaining the significance of this adaptation and its advantages over the original HiBSA algorithm would clarify its contribution.
>
> Thanks for the comment. The HiBSA is a algorithmic framework for the solution of min-max optimization problems which at each iteration solves an surrogate problem that is an approximation of the original one. MAT-SD is min-max algorithm specifically developed for the solution of the adversarial training problem (1). We can see MAT-SD as an implementation-friendly version of the HiBSA algorithm for the following reasons.
>
> - In HiBSA the gradient step over the inner variable $\delta$ is performed on a regularized version of the original objective. Specifically, the gradient step in the inner problem is performed over the problem $\widetilde{\mathcal{L}}(w^{r+1},\delta)-\gamma^{r}\|\delta\|^{2}$, where $\gamma^{r}$ is a diminishing regularization parameter, rather than directly on $\widetilde{\mathcal{L}}(w^{r+1},\delta)$. In MAT-SD, we omit the regularization term $-\gamma^{r}\|\delta\|^{2}$ because its inclusion adds complexity with the introduction of a new hyperparameter that needs to be tuned appropriately. We note that this does not affect negatively our approach as in our setting the magnitude of $\delta$ is small by construction, and hence the term $\gamma^{r}\|\delta\|^{2}$  takes very small values (assuming $\gamma^{r}$ is diminishing according to HiBSA specifications). Therefore, the contribution of the term $-\gamma^{r}\|\delta\|^{2}$ is insignificant.
> - In HiBSA the gradient update of the outer problem used an adaptive stepsize $\beta^{r}$. In MAT-SD we keep this stepsize fixed. Again, this does not diminish the utility of our method as in the theoretical analysis of HiBSA, the step size $\beta^{r}$ is chosen as a function of $\gamma^{r}$, which is no longer available in MAT-SD (as we dropped the term $-\gamma^{r}\|\delta\|^{2}$).
> - In general, the HiBSA framework can accommodate many different approximate surrogate objectives that can potentially result in complicated update rules. MAT-SD uses an approximation that results in a very simple iterative form. Specifically, MAT-SD's steps consist of a few gradient descent steps for the min problem to find a small perturbation $\delta_{att}$ , and then a single gradient ascent step per block for training the text encoder ($\theta$) and VAE decoder ($\phi$) block variables.
>
> We note that the above discussion is already present in the last paragraph of Section "3.4 The AT for SD, and the Proposed MAT-SD Algorithm".
>
> > Conclusion, limitations, and future work: -- The conclusion should concisely summarize the limitations of the proposed method to provide a balanced overview. -- Consider adding insights on the method's effectiveness and potential advantages/drawbacks when applied to other datasets. This would be a valuable addition to the results or following sections.
>
> Thanks for the feedback. In the revised version, we expanded the conclusion section to clearly summarize the main limitations of our work. The updated text is provided below.
> "In this study, we introduce an AT approach for SD. The AT method trains the system on a set of words, ensuring correct output even when the input undergoes certain adversarial modifications (e.g., "bicycle MJZM4").  However, our method has some limitations. First, it can robustify the model only against classes of attacks, specifically those attacks where the embeddings of the clean and adversarial prompts are close in a certain continuous text embedding space. Second, the applicability of the AT is limited to SD models. Therefore, in the future we plan to extend the applicability of the AT method to various text-to-image (T2I) systems beyond SD and to cover additional categories of adversarial attacks."

---

### Decision · Action_Editor_5oQJ · 2025-06-26

**Recommendation:** Reject

**Additional Comments:**

The proposed method is different from mainstream adversarial training approaches. While the reviewers agree that there are some merits in the current work, they share common concerns: there is major room for improvement in evaluation and analysis (classifier usage, missing ablation study, insufficient comparison to strong baselines and SOTA methods, limited improvement in performance, writing problem, generalizability). As a result, the current version of the paper is not ready for publication yet. We hope the reviews can help the authors to further revise the manuscript for a strong publication in the future.

**Audience:**

Yes

**Audience Explanation:**

The idea is based on adversarial training from the image classification model to solve a min-max optimization problem. The paper is easy to follow and well-written, except for some minor typos.

**Claims And Evidence:**

No

**Claims Explanation:**

This paper introduces an adversarial training procedure for Stable Diffusion. The authors conduct experiments to show that the proposed method enhances the model's robustness. However, the analysis done to evaluate their contribution are questionable.

**Resubmission Of Major Revision:**

The authors may consider submitting a major revision at a later time.